# High-entropy superparaelectrics with locally diverse ferroic distortion for high-capacitive energy storage

Jianhong Duan[1], Kun Wei[1], Qianbiao Du[1], Linzhao Ma[1], Huifen Yu[2], He Qi [2] ✉, Yangchun Tan[3], Gaokuo Zhong [3] ✉ & Hao Li [1] ✉

Superparaelectrics are considered promising candidate materials for achieving superior energy storage capabilities. However, due to the complicated local structural design, simultaneously achieving high recoverable energy density ($W_{rec}$) and energy storage efficiency ($\eta$) under high electric fields remains a challenge in bulk superparaelectrics. Here, we propose utilizing entropy engineering to disrupt long-range ferroic orders into local polymorphic distortion disorder with multiple $BO_6$ tilt types and diverse heterogeneous polarization configurations. This strategy reduces the switching barriers, thereby facilitating the emergence of superparaelectric behaviors with ideal polarization forms. Furthermore, it enables high polarization response, negligible remnant polarization, delayed polarization saturation, and enhanced breakdown electric fields ($E_b$) in high-entropy superparaelectrics. Consequently, an extraordinary $W_{rec}$ of 15.48 J cm$^{-3}$ and an ultrahigh $\eta$ of 90.02% are achieved at a high $E_b$ of 710 kV cm$^{-1}$, surpassing the comprehensive energy storage performance of previously reported bulk superparaelectrics. This work demonstrates that entropy engineering is a viable strategy for designing high-performance superparaelectrics.

With an increasing international focus on environmental protection, efficient energy storage technologies have become a focal point of societal concern[1–3]. Dielectric ceramic capacitors, with their ultrafast charge/discharge rate and ultrahigh power density, are extensively studied as a potential solution for energy storage[4–6]. However, the relatively low recoverable energy density ($W_{rec}$) and energy storage efficiency ($\eta$) of dielectric ceramic capacitors hinder their development towards miniaturization and integration[7–9]. Therefore, there is an urgent need to develop lead-free bulk ceramics with both ultrahigh $W_{rec}$ and $\eta$.

In recent years, numerous lead-free bulk ceramics have been developed for capacitive energy storage. For instance, high $W_{rec}$ of 11.4 J cm$^{-3}$ and 18.5 J cm$^{-3}$ have been realized in AgNbO$_3$ (AN)-based and NaNbO$_3$ (NN)-based antiferroelectric (AFE) ceramics, respectively[10,11]. Nevertheless, their $\eta$ values are capped at or below 80%, which is

mainly caused by the AFE-ferroelectric (FE) phase transition. Similar low $\eta$ phenomena are often observed in FEs or relaxor ferroelectrics (RFEs), such as K$_{0.5}$Na$_{0.5}$NbO$_3$ (KNN)-based, Bi$_{0.5}$K$_{0.5}$TiO$_3$ (BKT)-based, and BiFeO$_3$ (BF)-based ceramics[12–14]. Although a high $\eta$ (>90%) can be obtained in some linear dielectrics, such as CaTiO$_3$ (CT)-based and SrTiO$_3$ (ST)-based ceramics[15–18], the relatively low intrinsic polarization leads to their $W_{rec}$ usually being <7 J cm$^{-3}$. In addition, although an increase in the electric field contributes to an enhancement of $W_{rec}$, it is usually accompanied by an increase in remnant polarization ($P_r$), hysteresis losses, and leakage currents, all of which have negative impacts on $\eta$[7,19]. Therefore, the trade-off between $W_{rec}$ and $\eta$ has become a primary challenge in designing high-performance dielectric ceramics.

Recently, superparaelectrics (SPEs) developed in RFEs have been considered as promising candidate materials for energy storage[20–22].

[1]College of Electrical and Information Engineering, Hunan University, Changsha 410082, China. [2]Beijing Advanced Innovation Center for Materials Genome Engineering, Department of Physical Chemistry, University of Science and Technology Beijing, Beijing 100083, China. [3]Shenzhen Institute of Advanced Technology, Chinese Academy of Sciences, Shenzhen 518055, China. ✉e-mail: qiheustb@ustb.edu.cn; gk.zhong@siat.ac.cn; hli@hnu.edu.cn

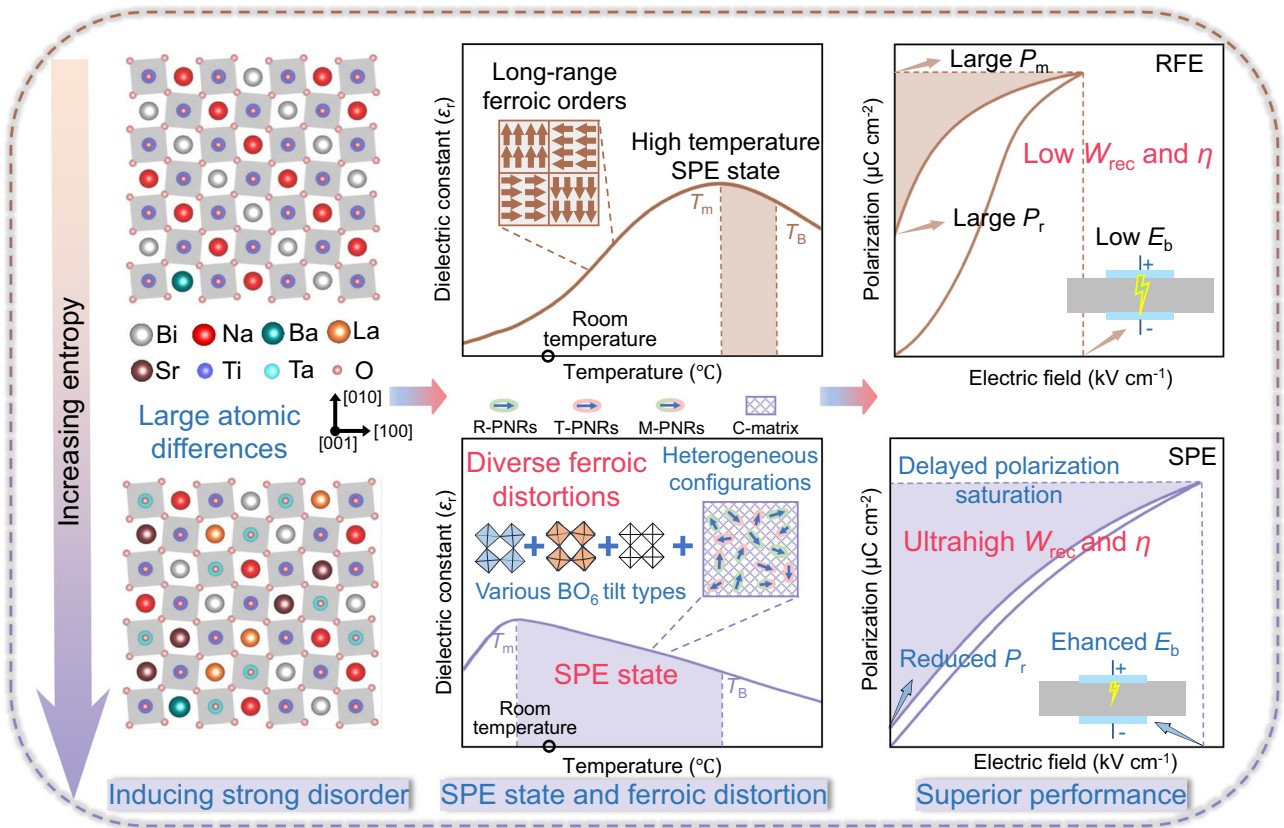

**Fig. 1 | Schematic diagram of using entropy engineering to achieve excellent comprehensive energy storage performance.**

The state of SPEs appears within the temperature range from $T_m$ (the temperature corresponding to the maximum dielectric constant) to $T_B$ (the Burns temperature) and is characterized by weakly coupled polar nanoregions (PNRs). Therefore, SPEs not only maintain a high maximum polarization ($P_m$) but also allow for flexible polarization redirection with small hysteresis, leading to a higher $\eta$ compared to conventional RFEs. Based on SPE engineering[23,24], $Bi_{0.5}Na_{0.5}TiO_3$ (BNT)-based ceramics achieved a $W_{rec}$ of 7.2 J cm$^{-3}$ and a $\eta$ of 86% at a breakdown electric field ($E_b$) of 430 kV cm$^{-1}$, and further, $BaTiO_3$-$Bi_{0.5}Na_{0.5}TiO_3$-$NaNbO_3$ (BT-BNT-NN) ternary ceramics reached a $W_{rec}$ of 10.59 J cm$^{-3}$ and a $\eta$ of 87.6% at a relatively high $E_b$ of 550 kV cm$^{-1}$. Despite advancements in researching SPEs, the challenge of simultaneously achieving ultrahigh energy density ($W_{rec} \geq 15$ J cm$^{-3}$) and minimal losses ($\eta \geq 90\%$) under high electric fields in bulk SPEs persists due to the lack of precise local structural design. From a thermodynamic perspective, when the long-range FE order in RFE materials shrinks down to nanodomains (or PNRs), the switching energy barriers also decrease. As the energy barrier diminishes to a level comparable to or lower than the thermal disturbance energy ($k_B T$, where $k_B$ represents the Boltzmann constant), nanodomains (or PNRs) can undergo flexible switching processes with minimal hysteresis, thus exhibiting ideal polarization form with SPE properties on the macroscopic scale[25,26]. In this sense, it should be feasible to design ideal SPEs by disrupting long-range FE order to construct flexible local polarization configurations, thereby lowering the switching energy barriers.

Here, we consider that entropy engineering has been demonstrated as an advanced strategy for regulating FE polarization in piezoelectric and energy storage dielectrics[27–30]. This is primarily due to the disordered component distribution leading to unmatched atomic size, mass, valence state, and electronegativity, which induce random local strain and electric fields, providing infinite possibilities for tuning the local polarization configurations. Therefore, we design high-entropy SPEs with superior comprehensive energy storage

performance through entropy engineering, as shown in Fig. 1. To ensure a large polarization response, $Bi_{0.47}Na_{0.47}Ba_{0.06}TiO_3$ (BNBT), known for its high polarization characteristics, is selected as the base material. The coexistence of tetragonal (T) and rhombohedral (R) phases in BNBT can reduce polarization anisotropy and promote polarization rotation, thereby lowering the switching energy barriers. Meanwhile, $Sr_{0.7}La_{0.2}Ta_{0.2}Ti_{0.75}O_3$ (SLTT) is added to BNBT to regulate configuration entropy ($S_{config}$) and create the $(1-x)$BNBT-$x$SLTT system (abbreviated as SLTT-$x$). The high-entropy effect, combined with the small $P_r$, the low hysteresis loss of $SrTiO_3$, and the large bandgap ($E_g$) of $La_2O_3$ (5.0 eV) and $Ta_2O_5$ (4.0 eV), further enhances the energy storage performance. Through entropy engineering, the long-range order is effectively disrupted, resulting in the formation of locally diverse ferroic distortions. These distortions encompass rich heterogeneous polarization configurations of R, T, and monoclinic (M)-like phases embedded in a cubic (C) matrix, as well as in-phase tilted, anti-phase tilted, and non-tilted $BO_6$ types. The induction of these heterogeneous configurations leads to the eventual realization of ideal SPE behavior, while the engineered multiple $BO_6$ tilt types effectively prevent premature polarization saturation. As expected, the high-entropy SPE (SLTT-0.30) exhibits a slender $P$-$E$ loop at a large $E_b$ of 710 kV cm$^{-1}$, yielding an impressive $W_{rec}$ of 15.48 J cm$^{-3}$ and an ultrahigh $\eta$ of 90.02%. These results signify a breakthrough in achieving superior energy storage capacities for bulk SPE ceramics.

## Results and discussion

### Induction of SPE state

As shown in Supplementary Table 1, the $S_{config}$ values of the SLTT-$x$ system are 0.88 $R$ ($x=0$), 1.47 $R$ ($x=0.20$), 1.54 $R$ ($x=0.25$), 1.61 $R$ ($x=0.30$), and 1.66 $R$ ($x=0.35$), respectively. Supplementary Fig. 1 illustrates the entropy-dependent dielectric properties. When $S_{config}$ reaches 1.61 (SLTT-0.30) and 1.66 $R$ (SLTT-0.35), the $T_m$ drops below room temperature, indicating that both samples have reached room

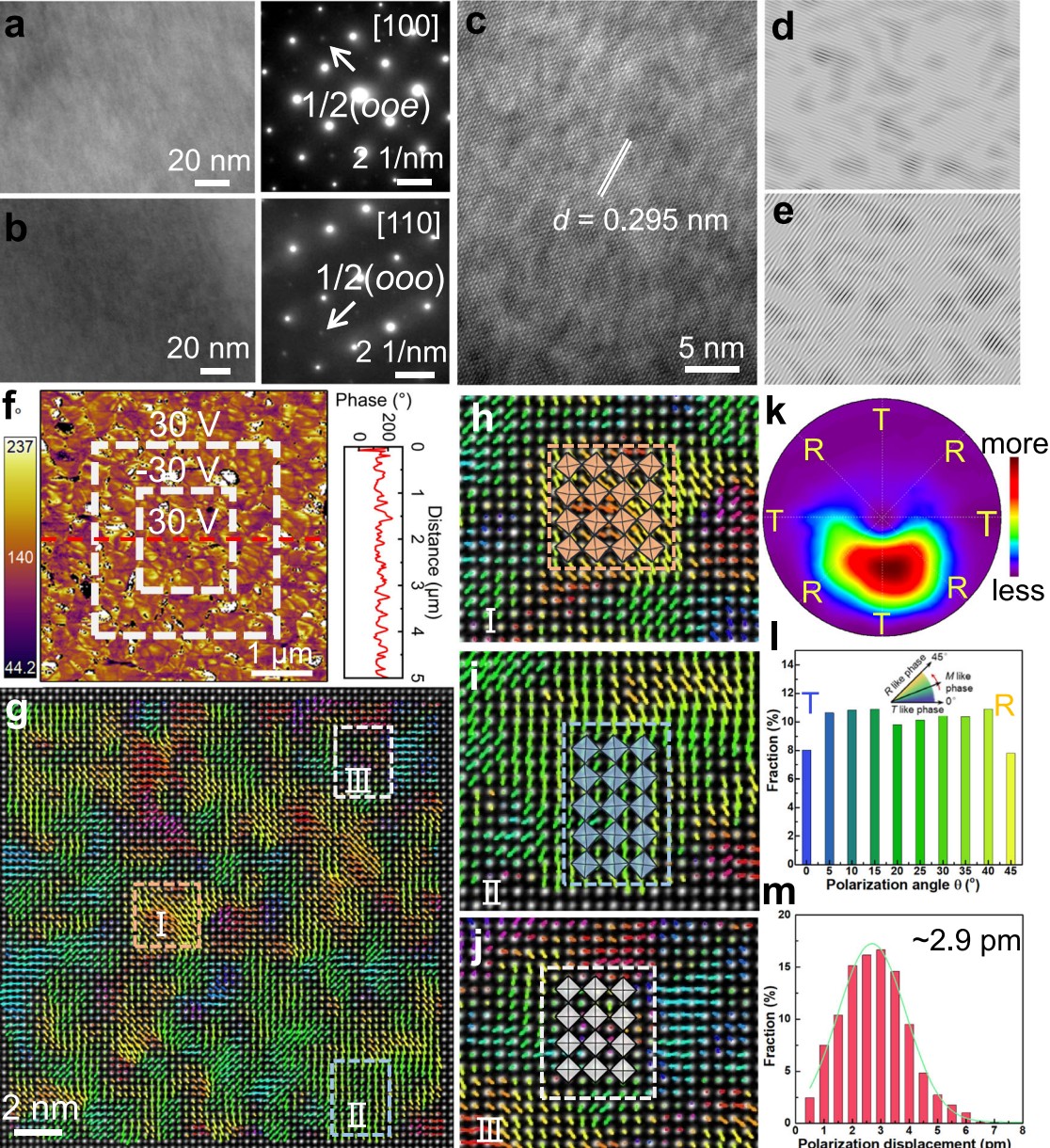

**Fig. 2 | Locally diverse ferroic distortion in SLTT-0.30 high-entropy SPE.** High-resolution TEM images and SAED patterns along (**a**) [100]$_c$ and (**b**) [110]$_c$. **c** High-resolution TEM image and (**d**, **e**) corresponding inverse fast Fourier transform patterns. **f** Out-of-plane PFM phase image after ± 30 V poling treatment and corresponding piezoresponse phase profile generated from the line scan. **g** HAADF-STEM polarization vector image recorded along [100]$_c$. **h–j** The magnified images of the labeled region in (**g**). The insets show the (**h**) anti-phase tilted, (**i**) in-phase tilted, and (**j**) non-tilted modes of the oxygen octahedron. **k** Statistical distribution of the amplitude and angle of the polarization vector. **l** Quantitative analysis of polarization angle. **m** Quantitative analysis of polarization amplitude.

temperature SPE states. Supplementary Fig. 2 shows the unipolar *P-E* loops of SLTT-*x* under low applied electric fields. As $S_{config}$ increases, the *P-E* loops become thinner, which signifies a transition from RFE to SPE. The overall energy storage performance under the same electric field is evaluated by the figure of merit $W_F = W_{rec}/(1 − \eta)$[31], as depicted in Supplementary Fig. 2c. SLTT-0.30 high-entropy SPE achieves the optimal $W_F$ of 44.17, which is attributed to its high $W_{rec}$ and $\eta$, thus highlighting its excellent potential for energy storage compared to SLTT-0.35.

**Induction of oxygen octahedral tilt**

X-ray diffraction (XRD) reveals that all SLTT-*x* ceramics possess a typical perovskite structure without impurities (Supplementary Fig. 3). The refinement results and phase compositions obtained by Rietveld refinement via GSAS-II software are shown in Supplementary Fig. 4[32].

SLTT-*x* ceramics exhibit a phase structure in which the R and T phases coexist. As *x* increases, the proportion of R phase decreases and the proportion of T phase increases. Supplementary Fig. 5 shows the Raman spectra of SLTT-*x* ceramics. Characteristic variations of the peaks related to cation disorder and BO$_6$ octahedral distortion are observed[33,34]. Moreover, double splitting is detected in both B−O bond (200−400 cm$^{-1}$) and BO$_6$ octahedral (400−700 cm$^{-1}$) vibrational modes, further indicating the coexistence of R phase and T phase in SLTT-*x* ceramics[35]. Figure 2a, b and Supplementary Fig. 6 show the transmission electron microscope (TEM) images and selected area electron diffraction (SAED) patterns of the SLTT-0.30 ceramic along the [100]$_c$ and [110]$_c$ directions. The absence of the large-scale ferroelectric domain in both the low-magnification and high-resolution TEM images points to the lack of both long-range polarization and BO$_6$ tilt order. The SAED patterns exhibit 1/2(*ooe*) type superlattice diffraction

dots caused by the in-phase tilt of the oxygen octahedron and 1/2(*ooo*) type superlattice diffraction points caused by the anti-phase tilt of the oxygen octahedron, respectively[36,37]. These results demonstrate that different localized $BO_6$ tilt types are induced in SLTT-0.30 ceramic, which can impede the appearance of electric field-induced texture domain states, thereby delaying polarization saturation[27].

## Modulation of heterogeneous configurations

High-resolution TEM image in Fig. 2c and the corresponding inverse Fourier transform patterns in Fig. 2d, e of SLTT-0.30 ceramics display pronounced local lattice distortions and PNRs[38]. The presence of these small-sized PNRs is further confirmed by piezoresponse force microscopy (PFM), as illustrated in Fig. 2f. The PFM image after poling treatment with ±30 V of SLTT-0.30 ceramic shows no domain switching and no abrupt change in the corresponding phase profile. This observation contrasts with the PFM images and phase profile of SLTT-0 ceramic (Supplementary Fig. 7), proving the formation of the highly dynamic PNRs in SLTT-0.30 ceramic[12,39,40]. In high-entropy systems, the introduction of foreign ions with different properties enhances the local random field, which can disrupt the long-range order into small-sized PNRs, a phenomenon that provides for lowering the switching energy barriers as well as modulating the polarization configuration in polymorphic phase coexistence systems. Moreover, the presence of highly dynamic PNRs suppresses the heat induced by polarization rotation and carrier migration in the electric field, which contributes to an improved $E_b$ and reduced energy loss.

In order to characterize the local structure in more detail and accuracy, high-resolution scanning TEM with high-angle annular dark-field imaging (HAADF-STEM) is performed on SLTT-0.30 ceramic. The displacement of the B (A) site cation from the four nearest neighboring A (B) site cation centers in the HAADF-STEM images can describe the local polarization[41]. The polarization vector mappings of the A-site and B-site atoms are shown in Supplementary Fig. 8a, b, respectively. Different colors and lengths of arrows are further used to represent the direction and magnitude of the polarization vector, as depicted in Fig. 2g. Inter-nested polar and non-polar clusters are observed. Magnified images of the marked regions in Fig. 2g show the R phase, T phase, and C phase regions, as well as the corresponding anti-phase tilt, in-phase tilt, and non-tilt models of the $BO_6$ octahedron, respectively (Fig. 2h–j). The C phase can also be identified from the dark blue region in the polarization amplitude mapping (Supplementary Fig. 8c), which not only reduces the internal stresses when applying an electric field, but also promotes rapid polarization recovery after unloading the electric field[42]. In addition, transition regions independent of the R, C, and T phases are detected, which can be further demonstrated in the statistical distribution of polarization vectors (Fig. 2k) and the polarization angle mapping (Supplementary Fig. 8d). Such transition regions are considered to be M-like symmetries with different vector angles and amplitudes[28,43,44]. The random distribution of local C-R-T-M-like phases indicates the decrease in polarization anisotropy and the existence of a strongly perturbed random field[27,45–47], which can effectively reduce the switching energy barriers and thereby lead to ideal macroscopic SPE behavior. To quantitatively study the distribution of polarization vectors, the polarization angle is transformed to a range from 0° (T phase) to 45° (R phase) based on the projection direction and unit cell symmetry, as displayed in Fig. 2l. The number of polarization vectors at each angle is almost equal, which not only reaffirms the weak polarization anisotropy but also further confirms the existence of M-like configurations with multiple vector angles. Furthermore, the weak ferroelectricity with a small average ferroelectric displacement (~2.9 pm) suggests that the polarization configurations exhibit weakly coupled characteristics, as illustrated in Fig. 2m.

The high-entropy effect leads to significant differences in atom size, mass, valence state, and electronegativity, amplifying local structure disorder and causing random local fields. This phenomenon disrupts long-range ferroic order into locally diverse ferroic distortion with multiple $BO_6$ tilt types and rich heterogeneous configurations. The $BO_6$ tilt types can hinder the formation of electric field-induced long-range polarization. Locally interconnected C-R-T-M-like phases can drastically reduce the polarization anisotropy, thereby reducing the switching barrier, resulting in a flatter switching pathway and minimizing the hysteresis loss in the SPEs. Moreover, diverse polarization configurations can also increase the polarization direction and intensity, providing a strong polarization response.

## Energy storage properties of high-entropy SPE

Figure 3a illustrates the slim unipolar *P-E* loops of SLTT-0.30 high-entropy SPE, indicating the insensitivity of the $P_r$ to the electric field. Remarkably, a high $P_m$ of 57.36 μC cm$^{-2}$ and a low $P_r$ of 2.52 μC cm$^{-2}$ are achieved at a large $E_b$ of 710 kV cm$^{-1}$. The realization of these energy storage-friendly parameters is attributed to the design of local ferroic distortion. The high $E_b$ is also associated with entropy-induced lattice distortion, small average grain size ($G_a$), clear and dense grain boundaries, wide $E_g$, and ultralow dielectric loss (tan$\delta$) (Supplementary Fig. 9)[48]. We further simulate the electric field distribution, electric potential distribution, and electric tree evolution of SLTT-0 and SLTT-0.30 samples under applied electric fields, respectively. As depicted in Supplementary Fig. 10, the SLTT-0.30 high-entropy SPE not only exhibits more uniform potential and electric field distributions but also more effectively hinders electric tree propagation compared to the SLTT-0 sample. This is associated with the uniform grain distribution, higher grain boundary density, and lower dielectric constant ($\varepsilon_r$) in the SLTT-0.30 sample[49,50]. The reliability of $E_b$ is verified using Weibull distribution analysis, as shown in Supplementary Fig. 11. The Weibull modulus ($\beta$) of the SLTT-0.3 sample is 19.8, indicating that the sample possesses high reliability and homogeneity[51]. The calculated $E_b$ is 725.6 kV cm$^{-1}$, which is very close to the $E_b$ obtained in the *P-E* loop. Figure 3b displays the increase of electric field from 100 kV cm$^{-1}$ to 710 kV cm$^{-1}$ with an approximately parabolic increase in both the total energy density ($W_{tot}$) and $W_{rec}$, while $\eta$ is consistently >90%. Consequently, the SLTT-0.30 high-entropy SPE achieves a "dual high" performance with $W_{rec}$ of ~15.48 J cm$^{-3}$ and $\eta$ of ~90.02%, which is a significant improvement in performance compared to the SLTT-0 ceramic ($S_{config}$ increased by 82%, $E_b$ increased by 610%, $W_{rec}$ increased by 1885%, and $\eta$ increased by 175%), as shown in Fig. 3c. Importantly, compared to reported bulk SPEs, RFEs, AFEs, high-entropy ceramics, and linear dielectrics, SLTT-0.30 high-entropy SPE exhibits a greater advantage in comprehensive energy storage performance (Fig. 3d and Supplementary Table 2). Furthermore, compared to these lead-free ceramics with $\eta$ > 90%, the SLTT-0.30 high-entropy SPE shows a higher $W_{rec}$ (Fig. 3e and Supplementary Table 2). These results suggest a breakthrough in overcoming the bottleneck of simultaneously achieving high $W_{rec}$ and low energy loss under high electric fields.

## Stability and charge-discharge performance of high-entropy SPE

In situ Raman spectra are measured to evaluate the temperature-dependent structural properties of SLTT-0.30 high-entropy SPE. As depicted in Fig. 4a, the deconvolution results indicate that the number of Raman peaks remains constant with increasing temperature, suggesting the local symmetry is maintained over a wide temperature range. The intensity of the peaks at ~291 cm$^{-1}$ and ~510 cm$^{-1}$ decreases, along with an increase in full width at half maximum (FWHM) and a gradual shift towards lower wavenumbers (Fig. 4b, c). This phenomenon demonstrates an increase in structural disorder, which helps suppress the increase of $P_r$ during a rise in temperature[52,53]. In situ XRD measurements from 25 °C to 250 °C are also used to investigate the stability of SLTT-0.30 ceramic. As shown in Fig. 4d, e, the (111) and (200) peaks both remain in their initial states without significant splitting or merging, proving the structural stability of SLTT-0.30

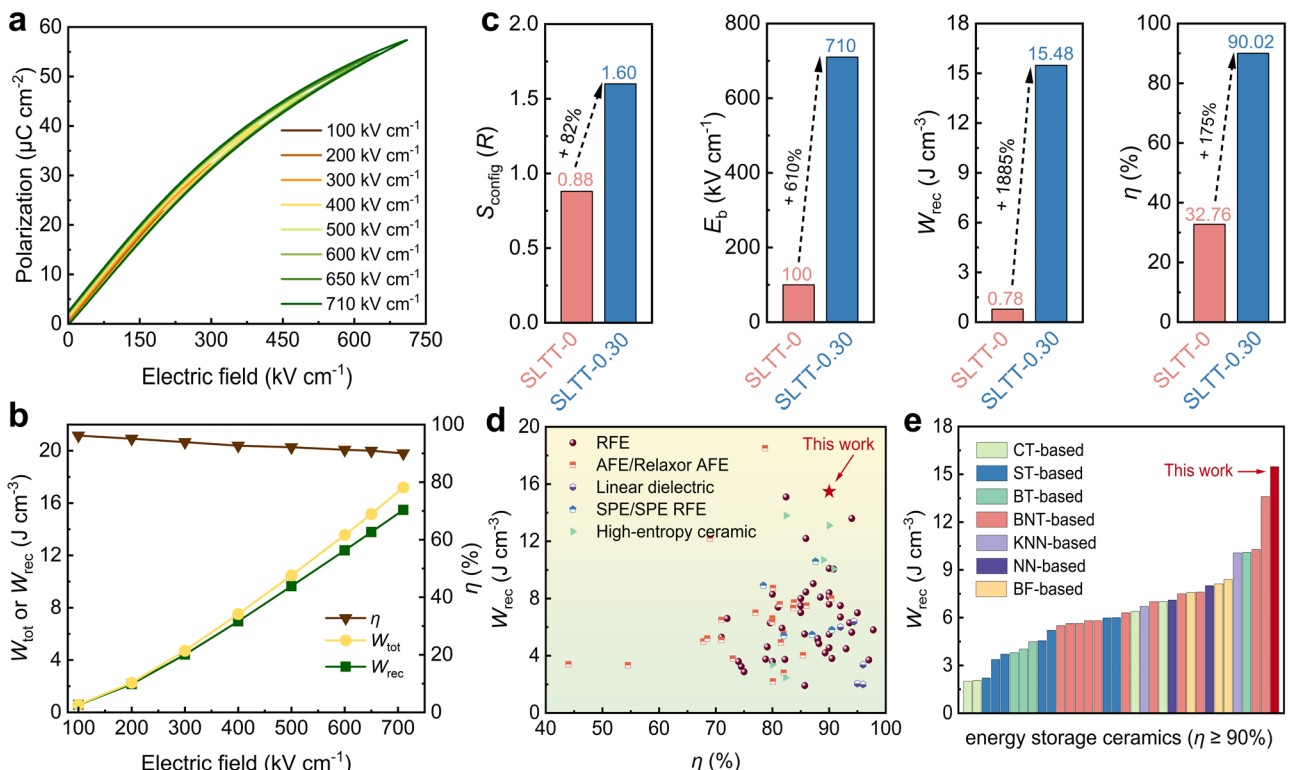

**Fig. 3 | Energy storage performance of high-entropy SPE. a** Electric field-dependent unipolar *P-E* loops of SLTT-0.30 ceramic. **b** Calculated $W_{rec}$, $W_{tot}$ and $\eta$ of SLTT-0.30 ceramic. **c** Comparison of $S_{config}$, $W_{rec}$, $\eta$, and $E_b$ between SLTT-0 and SLTT-0.30 ceramics. **d** Comparison of $W_{rec}$ and $\eta$ between SLTT-0.30 and reported lead-free bulk ceramics. **e** Comparison of $W_{rec}$ between SLTT-0.30 and reported lead-free bulk ceramics with a $\eta > 90\%$.

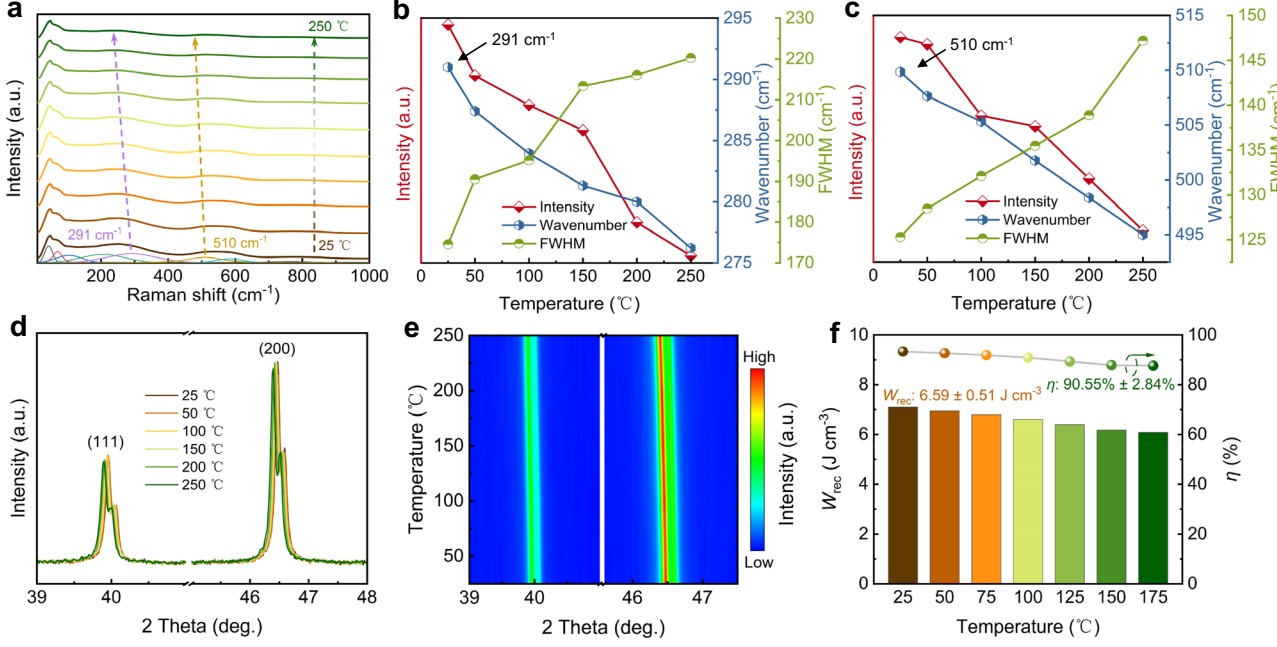

**Fig. 4 | Stability of high-entropy SPE. a** Temperature-dependent in situ Raman spectra of SLTT-0.30 ceramic. The variations in peak intensity, wavenumber, and FWHM of the peak at (**b**) 291 cm⁻¹ and (**c**) 510 cm⁻¹. **d** Temperature dependence of the (111) and (200) peaks in situ XRD of SLTT-0.30 ceramic. **e** Contour maps of the (111) and (200) peaks of SLTT-0.30 ceramic. **f** Temperature-dependent $W_{rec}$ and $\eta$ of SLTT-0.30 ceramic at 400 kV cm⁻¹.

ceramic[54]. As illustrated in Supplementary Fig. 12, the SLTT-0.30 ceramic is in the SPE state over a wide temperature range from $T_m$ of ~20.3 °C to $T_B$ of ~347.3 °C, which guarantees excellent temperature stability. As observed in Supplementary Fig. 13, the $\varepsilon_r$ and $\tan\delta$ of SLTT-0.30 ceramic fluctuate weakly with frequency, favoring the realization of frequency-insensitive energy storage performance. Furthermore, the entropy-induced polymorphic PNRs weaken the inter-coupling effect, lower the switching barrier, and promote the polarization rotation. Thus, these PNRs can respond quickly to the applied electric field, leading to enhanced stability[43,51]. Benefiting from these

synergistic effects, the energy storage performance of SLTT-0.30 ceramic demonstrates excellent stability at different temperatures and frequencies. As illustrated in Supplementary Fig. 14a, b, the unipolar $P$-$E$ loops maintain slim shapes at different temperatures and frequencies. Therefore, the SLTT-0.30 ceramic achieves superior temperature insensitivity over the temperature range of 25–175 °C, with $W_{rec} \approx 6.59 \pm 0.51\,J\,cm^{-3}$ and $\eta \approx 90.55 \pm 2.84\%$ (Fig. 4f). It also exhibits significant frequency insensitivity over the frequency range of 1–500 Hz, with $W_{rec} \approx 7.43 \pm 0.22\,J\,cm^{-3}$ and $\eta \approx 87.58 \pm 0.91\%$ (Supplementary Fig. 14c). These superior stabilities are superior to most reported high-performance lead-free bulk ceramics[32,34,55–58].

Supplementary Fig. 15 shows the excellent charge-discharge properties of the SLTT-0.30 high-entropy SPE. It achieves a high discharge energy density ($W_{dis} = 2.37\,J\,cm^{-3}$) in an ultrafast time ($t_{0.9} = 33\,ns$) under $320\,kV\,cm^{-1}$. The maximum current ($I_{max}$), current density ($C_D$), and power density ($P_D$) reach 19.5 A, 621.7 A cm$^{-2}$, and 93.3 MW cm$^{-3}$ under an electric field of 300 kV cm$^{-1}$, respectively. Additionally, the charge-discharge parameters demonstrate excellent temperature stability at 240 kV cm$^{-1}$. Minimal change in performance even at temperatures from 25 °C – 150 °C, with $W_{dis} \approx 1.42 \pm 0.05\,J\,cm^{-3}$, $t_{0.9} \approx 31 \pm 2\,ns$, $I_{max} \approx 13.37 \pm 0.02\,A$, $C_D \approx 425.56 \pm 0.78\,A\,cm^{-2}$, and $P_D \approx 46.83 \pm 0.09\,MW\,cm^{-3}$. Overall, the SLTT-0.30 ceramic shows great promise as a dielectric for energy storage capacitors due to its stability and impressive charge-discharge performance.

In summary, a lead-free bulk SPE has been developed using entropy engineering, demonstrating exceptional comprehensive energy storage performance with a large $W_{rec}$ of ~15.48 J cm$^{-3}$ and an ultrahigh $\eta$ of ~90.02% at 710 kV cm$^{-1}$. This performance represents the best reported to date for bulk SPEs. Moreover, the high-entropy SPE exhibits remarkable temperature stability, frequency stability, and superior charge-discharge performance. The impressive performance is attributed to the induction of locally diverse ferroic distortion by entropy engineering, including various $BO_6$ tilt types and heterogeneous symmetries, resulting in a room temperature SPE state with simultaneous lower $P_r$, improved $E_b$, and delayed polarization saturation. This work provides valuable insights into the interplay between entropy engineering, SPE behavior, local structure, and energy storage performance.

## Methods

### Sample preparation
The $(1 - x)$BNBT-$x$SLTT system (abbreviated as SLTT-$x$, $x = 0$, 0.20, 0.25, 0.30, 0.35) was fabricated by raw powders of $Bi_2O_3$ (purity of ≥99%), $Na_2CO_3$ (purity of 99.99%), $BaCO_3$ (purity of 99.8%), $TiO_2$ (purity of 99.5%), $SrCO_3$ (purity of 99.5%), $La_2O_3$ (purity of 99.9%), and $Ta_2O_5$ (purity of 99.5%) through a conventional solid-state reaction. The corresponding raw material powders were weighed according to the chemical formulas of BNBT and SLTT, with 1 mol% of $Bi_2O_3$ and $Na_2CO_3$ added to BNBT to prevent the volatilization of Bi and Na. The BNBT and SLTT raw material powders were dispersed for 18 h by ball milling with zirconia balls in ethanol, respectively, and then dried. The dried BNBT slurry was calcined at 850 °C for 5 h, while the dried SLTT slurry was calcined at 1000 °C for 5 h. Subsequently, the calcined powders of both materials were mixed and ball-milled for 20 h, followed by drying. The dried powder was mixed with polyvinyl alcohol (PVA) binder and pressed into pellets with a diameter of ~10 mm and a thickness of ~1 mm. The pellets were held at 600 °C for 3 h to volatilize PVA, then sintered at 1050–1180 °C for 2 h to obtain ceramic samples.

### Structural characterization
The phase structures of the ceramics at room temperature were examined by X-ray powder diffraction (XRD, Philips X'Pert Pro MPD, Netherlands). The surface microstructure of the ceramics after thinning, polishing, and thermal etching was analyzed using field emission

scanning electron microscopy (SEM, FEI, Quanta FEG250, USA). Selected area electron diffraction (SAED), domain morphology, and lattice fringes were examined by transmission electron microscopy (TEM, JEOL, JEM-2100, Japan). The response of domain structure to electric fields was characterized using piezoresponse force microscopy (PFM, Asylum Research, MFP-3D-Infinity, USA). The local structure of well-polished ceramics was analyzed by a Raman scattering spectrometer (Renishaw, inVia™, UK). To analyze temperature-dependent structural properties, Raman spectra were obtained using a Raman spectrometer (Horiba Jobin Yvon HR800, France) with a heating stage (Linkam, THM 600, UK) under 532 nm excitation, ranging from 25 °C to 250 °C, on the polished SLTT-0.30 ceramic samples. Additionally, temperature-dependent X-ray diffraction (XRD) of SLTT-0.30 ceramic was obtained using CuK$\alpha$ radiation at temperatures ranging from 25 °C to 250 °C with an XRD instrument (X'pert PRO, PANalytical, Netherlands). To analyze polarization vectors, amplitudes, and angles, high-angle annular dark-field (HAADF) atomic-scale images of the SLTT-0.30 ceramic were obtained using atomic-resolution STEM (aberration-corrected Titan Themis 3300), and custom MATLAB scripts were employed for analysis.

### Electrical performance measurement
For energy storage measurements, the samples were thinned and polished to ~0.05 ± 0.01 mm thickness, and gold electrodes with a diameter of 1.5 mm were prepared on their surfaces using ion sputtering. The ferroelectric analyzer (Aix ACCT, TF analyzer 1000, Germany) was utilized to measure the unipolar $P$-$E$ loops at room temperature and 10 Hz frequency. Additionally, the same equipment was employed to measure unipolar $P$-$E$ loops at different temperatures and frequencies to calculate the temperature-dependent and frequency-dependent energy storage performance. The dielectric properties were tested by a dielectric analysis instrument (Tongguo Technology, HCT1821, China). The charge-discharge performance of the ceramics were investigated using a charge-discharge tester (Tongguo Technology, CFD-003, China).

### Absorption spectrum
To obtain the bandgap of the samples, the absorption spectra ranging from 200 to 800 nm wavelength were measured using an ultraviolet-visible (UV-Vis) spectrophotometer (Cary 5000; Agilent, USA).

### Finite element simulation
Details of the finite element simulation are elaborated in the Supplementary information.

## Data availability
All data supporting this study and its findings are available within the article and its Supplementary Information. Any data deemed relevant are available from the corresponding author upon request.

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

## Acknowledgements

This work was supported by the National Natural Science Foundation of China (No.52102129, H.L.), the Hunan Provincial Natural Science Foundation of China (No.2023JJ30138, H.L.), the Science and Technology Innovation Program of Hunan Province (2023RC3094, H.L.), and the Shenzhen Science and Technology Program (Grant No. RCYX20200714114733204 and JCYJ20200109115219157, G.K.Z.).

## Author contributions

This work was conceived and designed by J.H.D., H.Q., G.K.Z., and H.L. Sample fabrication was performed by J.H.D. and K.W., who also conducted energy storage, dielectric, and charge-discharge performance tests, as well as analyzed relevant data. Finite element simulations of breakdown characteristics were conducted by H.F.Y. SEM and Raman spectroscopy were performed by H.L. and L.Z.M. UV-Vis data was collected and processed by H.L. PFM images were captured and processed by G.K.Z. HAADF-STEM imaging was carried out by H.Q. and processed accordingly. Temperature-dependent Raman and XRD data were collected by H.Q. and analyzed by J.H.D. XRD testing and corresponding data analysis were carried out by Y.C.T. The manuscript was drafted by J.H.D. and revised by H.L., Q.B.D., H.Q., G.K.Z., and Y.C.T. All authors participated in data analysis and discussions.

## Competing interests

The authors declare no competing interests
