## [Peer Review File · Nature Communications]

High-entropy superparaelectrics with locally diverse ferroic distortion for high-capacitive energy storageEditorial Note: Parts of this Peer Review File have been redacted as indicated to remove third-party material where no permission to publish could be obtained.

REVIEWER COMMENTS

Reviewer #1 (Remarks to the Author):

This manuscript reports on high-entropy SPEs with remarkable energy storage performance achieved through local structural design. The authors propose a strategy to design ideal SPEs using entropy engineering to tailor local polarization. A suite of micro-characterization techniques has been employed to reveal local polymorphic distortion with diverse BO₆ tilt types and heterogeneous polarization configurations brought on by high entropy effects, analyzing their positive impact on the SPE characteristics and energy storage performance. The manuscript provides a rigorous and in-depth analysis of the correlation between local structures and performance. From the perspectives of energy storage performance and novelty, it indeed holds substantial appeal and significance for the field of dielectrics.

Therefore, I think this manuscript can be accepted after addressing the following comments:

1. The design of local multi-phase symmetry is intriguing. Please elaborate on the advantages of this design over the common coexistence of the R and T phases in BNBT. In addition, what is the effect of distorted oxygen octahedra on energy storage performance?
2. For the designed composition of $(1 - x)\text{BNBT-xSLTT}$, why did the authors choose SLTT as the dopant? Besides improving the entropy of the BNBT material, are there any other reasons?
3. While in-depth exploration of the temperature stability of energy storage performance has been conducted, frequency stability is equally crucial for practical applications. Therefore, I suggest adding dielectric performance tests related to frequency in order to corroborate the frequency stability of the energy storage performance.
4. For the large E_b of 710 kV/cm obtained in the SLTT-0.30 sample, authors consider the high E_b is associated with entropy-induced lattice distortion, ultrafine grains (G_a), clear and dense grain boundaries, wide bandgaps (E_g), and ultralow dielectric loss ($\tan\delta$). Which factor may play a major role? Can you further explain?
5. Some minor issues:
 - (1) The four typical vibrational modes in Raman spectra should be indicated in Supplementary Fig. 3.
 - (2) The authors should present the configuration entropy of all the components at appropriate places to help the reader understand the relationship between SLTT and configuration entropy.
 - (3) Please provide the instrument parameters of the HAADF-STEM to ensure the results are perceived as reliable and convincing.
 - (4) There are errors in the reference list, such as incomplete information for Ref. 49.

Reviewer #2 (Remarks to the Author):

The current manuscript deals with a high entropy $(1 - x)\text{BNBT-xSLTT}$ SPE system. The manuscript reported simultaneous achievement of very high recoverable energy density ($W_{\text{rec}} = 15.48 \text{ J cm}^{-3}$) with ultrahigh efficiency ($\eta = 90.02\%$ under a high E_b of 710 kV cm^{-1}) by carefully engineering the local structural disorder. The obtained results are truly impressive. The volume, quality, and analysis of the experimental data presented in the manuscript are truly praiseworthy. The community will surely be benefitted from the research

results presented in the current manuscript, particularly the characterization and analysis of the local structural disorder to establish the room temperature stabilization of the SPE state. While going through the manuscript I was wondering what is the novelty presented in the current manuscript. After the reporting of SPE state by Pan et al. in the 1 October 2021 issue of Science (Science 374, 100-104 (2021)) and a perspective on “The superparaelectric battery” by Y-H Chu in the same issue, various energy storage research involving SPE state with remarkable comprehensive energy storage properties have been reported in the literature. High entropy design concept to achieve enhanced energy storage properties is also not new and well reported (For e.g., L Chen et al., Nature Communications, 2022, 13:3089 for thin film and M Zhang et al., Science 384, 185-189 (2024) for a MLCC structure). So, is it about reporting and very detailed characterizations of the locally diverse structural disorder of the ferroelectric state leading to excellent comprehensive energy storage properties of a relatively newer system, $(1 - x)\text{BNBT-xSLTT}$? In that case taking care of the following points would certainly make this manuscript more impactful and appealing to the potential readers.

1. A rationale behind choosing this particular composition/the design strategy of the investigated system would be useful.
2. How “x” in $(1 - x)\text{BNBT-xSLTT}$ was decided, any phase field/similar simulation studies were carried out to decide on this composition (x)? If it is only based on the configurational entropy (S), is there any reason for stopping at $x = 0.35$? Creating local chaos in the crystal structure, both in A and B sites of the perovskite structure, has been proven fruitful in developing high entropy ceramics. Any correlation between random local field (by varying the valency and size of the dopants) in A/B site with the polar cluster size leading to the SPE state? Some discussion in this line would be useful.
3. It is seen that the decrease in the polar cluster size in the RFE state is leading to the SPE state, any critical size (or its trigger, composition or temperature?) of this polar cluster that will lead to the SPE state? A discussion on this will be very helpful.
4. The manuscript claims an extraordinary energy storage property of a bulk SPE composition, but the electrical performance characterizations have been carried out on samples of thickness 0.05 ± 0.01 mm, can it really be called a bulk sample? I am not sure.
5. Are the sample thicknesses similar in the performance comparison of energy storage ceramics shown in figure 3 (d) and (e), if not, can they really be compared?
6. Is the electrical breakdown field, E_b , reported in the manuscript is the Weibull breakdown field? If not, better to report the Weibull E_b .

Reviewer #3 (Remarks to the Author):

The capacitive energy storage is in an area of current research and interest. Recently, more and more attention is paid to both ultrahigh W_{rec} and η , especially for the lead-free bulk ceramics. This work demonstrates that the $(1-x)\text{BNBT-xSLTT}$ presents a superior comprehensive energy storage performance due to the entropy engineering strategy, which disrupt long-range ferroic orders into local polymorphic distortion disorder and rich heterogeneous polarization configurations. The high-entropy superparaelectrics (SLTT-0.30) exhibits a large W_{rec} of 15.48 J cm^{-3} and an ultrahigh η of 90.02% at 710 kV cm^{-1} . The manuscript is well written, and the findings are important to the broader ferroelectrics community. However, there are some issues to be addressed. My comments and concerns are provided as follows:

- 1) In terms of “High-entropy”, it is a hot topic in recent. Does more element doping or solid

solution mean high entropy? Can the superparaelectrics only be attributed to the high-entropy systems?

2)The authors claim that the inverse relationship between W_{rec} and η in Introduction part. It is unclear where that comes from. Many factors are not certain to be related to the W_{rec} and η . The authors should give more comments. It would help reader understanding the relationship between W_{rec} and η .

3)It is not clear that high-entropy systems correspond to low free energy, potentially resulting in reduced barriers. Clarify in the text.

4)A key challenge of this work is that SLTT-0.30 is added into BNBT to regulate configuration entropy. This solid solution content is relatively large to the ceramic. How to ensure the absence of impurities and uniform distribution of multiple elements.

5)The authors use the HR-TEM results to demonstrate the absence of the large-scale ferroelectric domain. Did the authors check the lower-magnification TEM results? More TEM images should be provided in SI.

6)Fig. 2f shows the PFM image after poling treatment with ± 30 V of SLTT-0.30 ceramic and no domain switching was found. But why? Authors state that high-entropy may give rise to the low free energy, potentially resulting in reduced domain switching barriers. The pinning effect of defects maybe considered. Please elaborate this observation/result further.

Reviewer #4 (Remarks to the Author):

The paper reports on the fabrication of dielectric capacitors capable of simultaneously achieving high energy storage density and high charge-discharge efficiency by inducing an increase in configurational entropy in SPE states through the design of $(1-x)\text{BNBT}-x\text{SLTT}$ composition. The authors aptly demonstrate the existence of highly dynamic PNRs in T, R, M-like phases through TEM, PFM, and Raman analysis of sintered ceramics. Furthermore, they logically correlate the superior properties of high W_{rec} and efficiency in the SLTT-0.30 specimen with the presence of highly dynamic PNRs in various phases. Based on the excellent energy storage performance data, the paper appears to be quite well-written. However, from the perspective of this journal's requirements, the strategy of energy storage performance enhancement through entropy engineering is not entirely novel. While it's true that energy storage density and efficiency are important performance metrics for dielectric capacitors, it is now crucial to demonstrate the practical implementation of energy storage and utilization of dielectric capacitors. Therefore, I urge the authors to address and incorporate the following comments into the manuscript.

1. Practicality of Energy Density Values

In evaluating the energy density (W_{rec}) of the SLTT-0.30 specimen, the authors utilized samples with very thin thickness and small electrode areas. It cannot be guaranteed that high W_{rec} will be maintained as the electrode area and sample thickness increase. For instance, if the sample thickness is at the level of 0.3 to 0.5 mm and the electrode diameter is at the level of 3 to 5 mm, will high E_b and W_{rec} be sustained? (Indeed, many papers use thicknesses and electrode areas even larger than these.) It would be beneficial to demonstrate the behavior of E_b and W_{rec} concerning sample thickness.

2. Implementation of Energy Charge-Discharge Performance in Practical Devices

Building upon the preceding comment, it would be beneficial if the paper could introduce, even in a simple form, the implementation of a device utilizing the capacitor of SLTT-0.30 composition for energy charge-discharge applications. For instance, demonstrating the operation of a simple device driven by stored energy would be advantageous.

3. How was the E_b data in Fig. 3c determined? How many specimens were measured to determine E_b ? How is the reproducibility of the energy storage performance?

4. It would be beneficial to add the sources of the comparison data in Fig. 3d and e to the supplementary information.

5. For the fabrication of $(1-x)\text{BNBT}-x\text{SLTT}$ ceramics, the authors separately calcined BNBT and SLTT before mixing them again for sintering. Is there any specific reason for not simultaneously mixing all raw materials and calcining them? Are there no secondary phases present in the sintered $(1-x)\text{BNBT}-x\text{SLTT}$ ceramics? It would be beneficial to add XRD data and lattice information obtained through refinement of the sintered $(1-x)\text{BNBT}-x\text{SLTT}$ ceramics to the supplementary information.

6. Page 11, "Structural stability is also associated with reduced crystal structure symmetry caused by polymorphic PNRs." : If there could be further elaboration on this sentence, it would be beneficial.

7. In Fig. S8, the T_m of the SLTT-0.30 specimen appears to be higher than 50°C . This contradicts the results shown in Fig. S1d. Additionally, there is a typo "date" in Fig. S8 that needs to be corrected.

Response Letter to Manuscript “High-entropy superparaelectrics with locally diverse ferroic distortion for high-capacitive energy storage” (NCOMMS-24-22221A)

Response to Reviewer’s Comments

Thank you for your valuable time and effort invested in the peer review of our manuscript. The insightful and constructive comments have helped us improve the manuscript further. Based on these comments, we performed additional experiments, added new data, and made thorough revision on the manuscript. For your convenience, we have reproduced the original comments in black, provided point-by-point responses to comments in blue, and highlighted changes to the manuscript in yellow. We hope that the revisions can address your concerns. The point-by-point response is as follows:

Reviewer #1

Remarks to the Author: This manuscript reports on high-entropy SPEs with remarkable energy storage performance achieved through local structural design. The authors propose a strategy to design ideal SPEs using entropy engineering to tailor local polarization. A suite of micro-characterization techniques has been employed to reveal local polymorphic distortion with diverse BO_6 tilt types and heterogeneous polarization configurations brought on by high entropy effects, analyzing their positive impact on the SPE characteristics and energy storage performance. The manuscript provides a rigorous and in-depth analysis of the correlation between local structures and performance. From the perspectives of energy storage performance and novelty, it indeed holds substantial appeal and significance for the field of dielectrics. Therefore, I think this manuscript can be accepted after addressing the following comments:

Response: We sincerely appreciate your positive comments and recognition of our work. We designed diverse ferroic distortions through entropy engineering, including rich heterogeneous polarization configurations and multiple BO_6 tilt types. This design

significantly lowers the switching energy barrier and makes the switching path flatter, enabling SPE behavior with ideal polarization forms. Therefore, we achieved an ultrahigh W_{rec} of 15.48 J cm^{-3} and an efficiency of over 90% in high-entropy bulk SPE. Importantly, based on your valuable suggestions, we have made corresponding revisions to the manuscript, which have significantly improved its quality.

Comment 1: The design of local multi-phase symmetry is intriguing. Please elaborate on the advantages of this design over the common coexistence of the R and T phases in BNBT. In addition, what is the effect of distorted oxygen octahedra on energy storage performance?

Response: Thank you for your comments. We have designed the local structure of rhombohedral (R), tetragonal (T), and monoclinic (M)-like polarization configurations nested with cubic (C) phase nonpolar regions by entropy engineering. Compared with conventional two-phase coexistence, this multiple symmetry structure can further attenuate both in polarization anisotropy and switching energy barrier. Consequently, the electric field-induced polarization rotation is smoother, leading to an SPE behavior with an ideal polarization form on the macroscopic scale^{[Energy Environ. Sci. 16, 4511–4521 (2023)], [Appl. Phys. Lett. 124, 090501 (2024)], [Science 384, 185–189 (2024)]}. The local C phase is a non-polar phase, which can inhibit the polarization rotation and reduce the internal stress when loading the electric field, and promote the polarization recovery after unloading the electric field, so it can achieve the purpose of delaying polarization saturation and reducing energy loss^[Adv. Mater. 36, 2313285 (2024)]. Both the R and M-like polarizations possess high polarization strengths, which are beneficial for the ceramic to achieve large polarization responses^[Nano Energy 112, 108458(2023)]. Additionally, the M-like polarization has different polarization directions, which can further enhance the flexibility of the polarization configuration and lower switching barriers, thus facilitating the realization of polarization form with SPE behavior^{[Nano Energy 104, (2022) 107910], [Acta Mater. 236, 118115 (2022)]}.

In this work, the disordered oxygen octahedra have a positive effect on the energy storage performance. Random oxygen octahedral distortions in ceramics can absorb

part of the electrical energy from the applied electric field, hindering the formation of textured domain states under the electric field and thus favoring delayed polarization saturation^[Nat. Commun. 13, 3089 (2022)]. Moreover, this local disorder can further disrupt long-range ordering, leading to enhanced relaxor behavior. Finally, systems with oxygen octahedral distortions typically exhibit excellent stability^[InfoMat 5, 12488 (2023)].

Comment 2: For the designed composition of $(1-x)\text{BNBT}-x\text{SLTT}$, why did the authors choose SLTT as the dopant? Besides improving the entropy of the BNBT material, are there any other reasons?

Response: The selection of $\text{Sr}_{0.7}\text{La}_{0.2}\text{Ta}_{0.2}\text{Ti}_{0.75}\text{O}_3$ (SLTT) as a dopant in the $(1-x)\text{BNBT}-x\text{SLTT}$ system has been considered from the following aspects, besides increasing entropy:

1) Derivatives related to SLTT, such as $\text{Sr}_{0.7}\text{La}_{0.2}\text{TiO}_3$, $\text{Sr}(\text{Al}_{0.5}\text{Ta}_{0.5})\text{O}_3$, $\text{Sr}_{0.7}\text{La}_{0.2}\text{Zr}_{0.15}\text{Ti}_{0.85}\text{O}_3$, etc., can better optimize the storage performance of BNT-based ceramics, providing a basis for selecting SLTT as a modifier^{[Ceram. Int. 50, 5021–5031 (2024)], [Small 19, 2206958 (2022)], [Chem. Eng. J. 446, 137105 (2022)]}.

2) In SLTT, SrTiO_3 has a very low loss, P_r , and polarization hysteresis, which is beneficial to reduce the hysteresis loss of $\text{SLTT}-x$ and P_r . In addition, its C phase plays a key role in delaying polarization saturation and reducing energy loss.

3) In SLTT, SrTiO_3 (3.4 eV), La_2O_3 (5.0 eV) and Ta_2O_5 (4.0 eV) have a wide intrinsic bandgap (E_g), which can be adjusted to enhance the conductivity and reduce the leakage current, which is essential for improving E_b and η .

4) The substitution of volatile cations (Bi^{3+} and Na^+) by (Sr^{2+} and La^{3+}) results in oxygen vacancy concentrations decreasing, which can inhibit the generation of holes as well as enhance densification^[Chem. Eng. J. 446, 137105 (2022)]. The refractory nature of Ta can inhibit grain growth. Moreover, the introduction of multiple ions can increase the lattice distortion and thus increase the lattice strain energy, which can inhibit the grain growth^{[Nat. Mater. 21, 1074–1080 (2022)], [Science 384, 185–189 (2024)]}.

Comment 3: While in-depth exploration of the temperature stability of energy storage

performance has been conducted, frequency stability is equally crucial for practical applications. Therefore, I suggest adding dielectric performance tests related to frequency in order to corroborate the frequency stability of the energy storage performance.

Response: Thank you very much for your suggestion. Frequency stability is indeed an important factor affecting practical applications. To explore this, we added frequency-dependent dielectric property tests according to your suggestion. As shown in Supplementary Fig. 13, it can be observed that ϵ_r and $\tan\delta$ have little dependence on frequency, which provides important support for the frequency stability of SLTT-0.30 ceramics in energy storage performance. Therefore, we added this content to the manuscript and Supplementary Information: As can be observed in Supplementary Fig. 13, the ϵ_r and $\tan\delta$ of SLTT-0.30 ceramic fluctuate weakly with frequency, favoring the realization of frequency-insensitive energy storage performance.

Supplementary Fig. 13 Frequency dependence of ϵ_r and $\tan\delta$ of the SLTT-0.30.

Comment 4: For the large E_b of 710 kV/cm obtained in the SLTT-0.30 sample, authors consider the high E_b is associated with entropy-induced lattice distortion, ultrafine grains (G_a), clear and dense grain boundaries, wide bandgaps (E_g), and ultralow dielectric loss ($\tan\delta$). Which factor may play a major role? Can you further explain?

Response: Thank you for your question. We believe that these factors all have an impact on E_b , and they are inextricably linked to each other. Judging by the mechanism of their impact on E_b , the lattice distortion has emerged as one of the most important factors. The corresponding analysis is as follows:

- 1) Bandgap (E_g): The larger E_g means that it is difficult for electrons to transition

from the valence band to the conduction band, leading to a larger intrinsic resistivity and E_b . Therefore, E_g and E_b have the following relationship with each other [ACS Appl. Mater. Interfaces 13, 51218–51229 (2021)].

$$E_b = 1.36 \times 10^7 \times \left(\frac{E_g}{4.0}\right)^3 (V/cm).$$

2) Dielectric loss ($\tan\delta$): Low $\tan\delta$ indicates reduced energy dissipation during charging and discharging, which reduces heat generation and therefore reduces the probability of thermal breakdown.

3) Grains and grain boundaries: Smaller grain sizes favor the enhancement of E_b . On the one hand, ceramics with smaller grain sizes imply more grain boundaries. The depleted space charge layer exists at the grain boundary, which is analogous to a Schottky barrier at a semiconductor interface that can impede the migration of charge carriers [ACS Appl. Mater. Interfaces 13, 51218–51229 (2021)]. On the other hand, ceramics with smaller grain sizes have a denser internal structure with fewer voids, which can reduce harmful discharge concentrations, local failure probabilities, and dielectric losses [Chem. Eng. J. 429, 132165 (2022)], [InfoMat 5, 12488 (2023)].

4) Lattice distortion: Lattice distortion can increase the probability of electron collisions with lattice atoms, which enhances electron scattering, leading to a decrease in conductivity and an increase in E_b [Nat. Energy 8, 956-964 (2023)]. In high-entropy ceramics, the presence of strong lattice distortion usually exhibits microstructural features of grain refinement. This is mainly due to the increase in lattice strain energy caused by lattice distortion, which hinders grain growth [Nat. Mater. 21, 1074–1080 (2022)], [Science 384, 185–189 (2024)]. Furthermore, lattice distortion can lead to solid solution hardening, and thus high entropy materials are able to withstand the compressive forces generated by the electrostatic attraction of the surface charges and stress generated by the electrostriction effect while reducing the possibility of electromechanical breakdown [InfoMat 5, 12488 (2023)], [Adv. Mater. 36, 2305453 (2024)].

It can be found that lattice distortion not only promotes E_b from a microscopic point of view but also affects grain growth, which indirectly has an effect on E_b . In addition, lattice distortion is a structural feature that can be directly observed by

advanced characterization means, which provides a solid foundation for studying the relationship between structure and performance from a deeper perspective. Therefore, we think the most important factor determining E_b is lattice distortion.

Comment 5: Some minor issues:

1) The four typical vibrational modes in Raman spectra should be indicated in Supplementary Fig. 3.

2) The authors should present the configuration entropy of all the components at appropriate places to help the reader understand the relationship between SLTT and configuration entropy.

3) Please provide the instrument parameters of the HAADF-STEM to ensure the results are perceived as reliable and convincing.

4) There are errors in the reference list, such as incomplete information for Ref. 49.

Response: Thank you very much for your suggestions. We have made revisions accordingly.

1) We have labeled four typical vibrational modes in the Raman spectra.

Supplementary Fig. 5 Raman spectra of the SLTT- x ceramics.

2) We have added a description of the configuration entropy of each component on page 5 of the manuscript: As shown in Supplementary Table 1, the S_{config} of SLTT- x systems are $0.88R$ ($x = 0$), $1.47R$ ($x = 0.20$), $1.54R$ ($x = 0.25$), $1.61R$ ($x = 0.30$), and $1.66R$ ($x = 0.35$), respectively.

3) We have supplemented the instrument parameters for HAADF-STEM in the **Structural characterization** section.

4) We have checked all the references and made the following modifications to the incorrect references:

33. Li, D. et al. Lead-free relaxor ferroelectric ceramics with ultrahigh energy storage densities via polymorphic polar nanoregions design. *Small* **19**, 2206958 (2022).
38. Ma, Q. et al. Excellent energy-storage performance in lead-free capacitors with highly dynamic polarization heterogeneous nanoregions. *Small* **19**, 2303768 (2023).
44. Zhang, Y. et al. High-performance ferroelectric based materials via high-entropy strategy: design, properties, and mechanism. *InfoMat* **5**, 12488 (2023).
48. Zhao, W. et al. Broad-high operating temperature range and enhanced energy storage performances in lead-free ferroelectrics. *Nat. Commun.* **14**, 5725 (2023).
49. Zhang, Y. et al. Superior energy-storage properties in Bi_{0.5}Na_{0.5}TiO₃-based lead-free ceramics via simultaneously manipulating multiscale structure and field-induced structure transition. *ACS Appl. Mater. Interfaces* **14**, 40043–40051 (2022).

Reviewer #2

Remarks to the Author: The current manuscript deals with a high entropy $(1 - x)\text{BNBT-}x\text{SLTT}$ SPE system. The manuscript reported simultaneous achievement of very high recoverable energy density ($W_{\text{rec}} = 15.48 \text{ J cm}^{-3}$) with ultrahigh efficiency ($\eta = 90.02\%$ under a high E_b of 710 kV cm^{-1}) by carefully engineering the local structural disorder. The obtained results are truly impressive. The volume, quality, and analysis of the experimental data presented in the manuscript are truly praiseworthy. The community will surely be benefitted from the research results presented in the current manuscript, particularly the characterization and analysis of the local structural disorder to establish the room temperature stabilization of the SPE state.

While going through the manuscript I was wondering what is the novelty presented in the current manuscript. After the reporting of SPE state by Pan et al. in the 1 October 2021 issue of Science (Science 374, 100-104 (2021)) and a perspective on “The superparaelectric battery” by Y-H Chu in the same issue, various energy storage research involving SPE state with remarkable comprehensive energy storage properties have been reported in the literature. High entropy design concept to achieve enhanced energy storage properties is also not new and well reported (For e.g., L Chen et al., Nature Communications, 2022, 13:3089 for thin film and M Zhang et al., Science 384, 185-189 (2024) for a MLCC structure). So, is it about reporting and very detailed characterizations of the locally diverse structural disorder of the ferroelectric state leading to excellent comprehensive energy storage properties of a relatively newer system, $(1 - x)\text{BNBT-}x\text{SLTT}$? In that case taking care of the following points would certainly make this manuscript more impactful and appealing to the potential readers.

Response: We sincerely thank you for recognizing our work and pointing out its significance and value. We are very encouraged by this. As you mentioned, our work is to use detailed characterization to prove the contribution of the locally diverse structural disorder to excellent comprehensive energy storage performance in the innovative SPE system $(1 - x)\text{BNBT-}x\text{SLTT}$. The choice of BNBT is mainly due to its high P_m and multi-phase characteristics, which can provide unlimited possibilities for improving

energy storage performance. For the selection of SLTT, we considered its regulation of configurational entropy, its own polarization characteristics (small P_r and hysteresis loss), high bandgap, etc. Our explanation for this important innovation of combining locally diverse ferroic distortion with SPE state through entropy engineering is as follows:

Indeed, both Pan et al. and Chu have significantly advanced our understanding of SPE films, while works by Chen et al. and Zhang et al. have contributed profoundly to high-entropy design research. Building upon these foundational studies, we have found a link between structural disorder caused by high entropy and the thermodynamics of the SPE state. Structural disorder due to high entropy usually disrupts the long-range ordering into small-size PNRs, which provides infinite possibilities for the emergence of diverse localized polarization configurations, especially in multiphase systems such as BNBT. From a thermodynamic point of view, the design of an ideal SPE requires low domain switching energy barriers for polarization reorientation with small hysteresis. Therefore, a connection between high entropy and SPE can be expected since the switching energy barrier decreases with decreasing domain size and polarization anisotropy. Based on this, we have designed locally diverse ferroic distortions by entropy engineering, which reducing the domain size and polarization anisotropy, and inducing different BO_6 tilt types. This phenomenon successfully induces an ideal SPE state with small P_r and hysteresis, large E_b , and delayed polarization saturation.

The above-mentioned reports on SPE mainly focus on the study of dielectric films. Zhang et al.'s report on high-entropy focuses on the study of multilayer ceramic capacitors (MLCCs) with a W_{rec} of 20.8 J cm^{-3} and an η of 97.5%, while the W_{rec} of the single-layer bulk ceramic is 10 cm^{-3} and η is 93.5%. The W_{rec} of the high-entropy bulk ceramic prepared by Chen et al. is 10.06 J cm^{-3} and η is 90.8%. In this work, we demonstrate that entropy engineering is an advanced approach to designing ideal bulk SPEs through advanced characterization. This design induces local ferroic distortion, which reduces the switching barrier and polarization anisotropy, and leads to excellent energy storage performance of $W_{\text{rec}} \sim 15.48 \text{ J cm}^{-3}$ and $\eta \sim 90.02\%$ in high-entropy bulk

SPEs. Through our in-depth analysis, we believe that this can provide a valuable basis for improving energy storage performance and is worth promoting.

Comment 1: A rationale behind choosing this particular composition/the design strategy of the investigated system would be useful.

Response: Thank you for your suggestion. The selection of matrix materials is particularly important for energy storage performance. We chose BNBT as the matrix material mainly because it can provide a large polarization response (high P_m), which is very important for W_{rec} . In addition, BNBT is shown to be characterized by the coexistence of R and T phases^[Chem. Eng. J. 420, 130475 (2021)]. Therefore, the polarization anisotropy can be reduced by inducing locally diverse polarization configurations to flatten the switching path and thus enhance the energy storage performance.^[Science 384, 185–189 (2024)] When selecting SLTT modifiers, in addition to the modulation of configuration entropy, the following main reasons were considered:

1) Derivatives related to SLTT, such as $Sr_{0.7}La_{0.2}TiO_3$, $Sr(Al_{0.5}Ta_{0.5})O_3$, $Sr_{0.7}La_{0.2}Zr_{0.15}Ti_{0.85}O_3$, etc., can better optimize the storage performance of BNT-based ceramics, providing a basis for selecting SLTT as a modifier^{[Ceram. Int. 50, 5021–5031 (2024)], [Small 19, 2206958 (2022)], [Chem. Eng. J. 446, 137105 (2022)]}.

2) In SLTT, $SrTiO_3$ has a very low loss, P_r , and polarization hysteresis, which is beneficial to reduce the hysteresis loss of SLTT- x and P_r . In addition, its C phase plays a key role in delaying polarization saturation and reducing energy loss.

3) In SLTT, $SrTiO_3$ (3.4 eV), La_2O_3 (5.0 eV) and Ta_2O_5 (4.0 eV) have a wide intrinsic bandgap (E_g), which can be adjusted to enhance the conductivity and reduce the leakage current, which is essential for improving E_b and η .

4) The substitution of volatile cations (Bi^{3+} and Na^+) by (Sr^{2+} and La^{3+}) results in oxygen vacancy concentrations decreasing, which can inhibit the generation of holes as well as enhance densification^[Chem. Eng. J. 446, 137105 (2022)]. The refractory nature of Ta can inhibit grain growth. Moreover, the introduction of multiple ions can increase the lattice distortion and thus increase the lattice strain energy, which can inhibit the grain growth^{[Nat. Mater. 21, 1074–1080 (2022)], [Science 384, 185–189 (2024)]}.

As suggested, we have updated the rationale for choosing this particular composition/design strategy for the survey system on page 4: The sentence “ $\text{Bi}_{0.47}\text{Na}_{0.47}\text{Ba}_{0.06}\text{TiO}_3$ (BNBT) with high polarization genes is selected as the base material. $\text{Sr}_{0.7}\text{La}_{0.2}\text{Ta}_{0.2}\text{Ti}_{0.75}\text{O}_3$ (SLTT) is added to it to regulate configuration entropy (S_{config}) and create the $(1-x)\text{BNBT}-x\text{SLTT}$ system (abbreviated as SLTT- x)” on page 4 has been updated to “To ensure a large polarization response, $\text{Bi}_{0.47}\text{Na}_{0.47}\text{Ba}_{0.06}\text{TiO}_3$ (BNBT), known for its high polarization characteristics, is selected as the base material. The coexistence of tetragonal (T) and rhombohedral (R) phases in BNBT can reduce polarization anisotropy and promote polarization rotation, thereby lowering the switching energy barrier. Meanwhile, $\text{Sr}_{0.7}\text{La}_{0.2}\text{Ta}_{0.2}\text{Ti}_{0.75}\text{O}_3$ (SLTT) is added to it to regulate configuration entropy (S_{config}) and create the $(1-x)\text{BNBT}-x\text{SLTT}$ system (abbreviated as SLTT- x). The high-entropy effect, combined with the small P_r , low hysteresis loss characteristics of SrTiO_3 and the high bandgap (E_g) characteristics of La_2O_3 (5.0 eV) and Ta_2O_5 (4.0 eV), further enhances the energy storage performance”.

Comment 2: How “ x ” in $(1-x)\text{BNBT}-x\text{SLTT}$ was decided, any phase field/similar simulation studies were carried out to decide on this composition (x)? If it is only based on the configurational entropy (S), is there any reason for stopping at $x = 0.35$? Creating local chaos in the crystal structure, both in A and B sites of the perovskite structure, has been proven fruitful in developing high entropy ceramics. Any correlation between random local field (by varying the valency and size of the dopants) in A/B site with the polar cluster size leading to the SPE state? Some discussion in this line would be useful.

Response: Thank you for your comments. Our decision on “ x ” was guided by configuration entropy (S_{config}) and a large amount of research data such as dielectric properties, microscopic morphology, field-induced polarization, and energy storage properties. Firstly, as shown in Table R1, the ceramics undergo a transition from low-entropy to high-entropy as well as RFE to SPE as x increases. The low-entropy $x = 0$ component, the medium-entropy $x = 0.20$ component, and the high-entropy $x = 0.25$ component are typical RFE states. At $x = 0.30$ and 0.35 , both high-entropy samples are SPEs. This phenomenon can already be well illustrated by the entropy-driven RFE to

SPE transition, which fits the design core of our study. Secondly, the P - E loops in Supplementary Fig. 2 show that the P - E loops for $x = 0.20$ and 0.25 have large P_r and hysteresis unfavorable for energy storage. As x increases, the P - E loops become thinner, and a very thin P - E loop has been obtained at $x = 0.35$. We calculated the W_{rec} , η , and W_{F} for $x = 0.20, 0.25, 0.30,$ and 0.35 at 200 kV cm^{-1} as shown in Table R2. It can be noticed that W_{rec} decreases from 2.15 to 1.65 when $x = 0.30$ is increased to $x = 0.35$, while η increases only slightly from 95.11% to 95.68%, which leads to a decrease in W_{F} from 44.17 to 38.16. From these trends, it can be hypothesized that setting x to 0.15 or 0.40 is detrimental to energy storage. Finally, from Supplementary Fig. 9, we can find that $x = 0.30$ possesses the densest grain distribution, the smallest G_a and the largest E_g , which are conducive to the enhancement of E_b . Based on the above discussion, increasing the SLTT content to $x = 0.35$ did not further improve the performance. We hypothesize that this may be due to the fact that S_{config} reaches an optimal level at $x = 0.30$, which is critical for improving energy storage performance. Therefore, we decided to set the SLTT content between 0.20 and 0.35.

Table R1 S_{config} , ceramic properties, and dielectric state of SLTT- x .

x	0	0.20	0.25	0.30	0.35
S_{config}	0.88R	1.47R	1.54R	1.61R	1.66R
Property	Low-entropy	Medium-entropy	High-entropy	High-entropy	High-entropy
State	RFE	RFE	RFE	SPE	SPE

Table R2 W_{rec} , η , and W_{F} of SLTT- x .

x	0.20	0.25	0.30	0.35
W_{rec}	2.58	2.33	2.15	1.65
η	78.22	85.22	95.11	95.68
W_{F}	11.83	15.78	44.17	38.16

In perovskites, developing high-entropy ceramics by introducing a large number of ions to induce local disorder has been shown to be very effective. And there is indeed a non-negligible correlation between random local fields and polar clusters, especially

in high-entropy systems. In high-entropy materials, unmatched atomic size, valence state, mass, and electronegativity can arise due to the occupation of ions with different properties in the A and B sites. This situation generates random local strain and electric fields, which can disrupt the conventional long-range ferroelectric order into weakly coupled polar nanoscale domains (PNRs). Therefore, the random field increases with S_{config} , which leads to an increase in the content and a decrease in the size of the PNRs, which will further reduce the switching barriers, driving the RFE state evolution to the SPE state^{[Nat. Energy 8, 956–964 (2023)], [InfoMat 5, 12488 (2023)], [Nano-Micro Lett. 15, 65 (2023)], [Nat. Commun. 13, 3089 (2022)]}. Accordingly, the introduction of chaos into the perovskite structure through a high-entropy strategy induces local random fields, which is an important factor in tuning the SPE state. To further enhance the quality of the manuscript, we have added a discussion of random fields to the manuscript.

1) The sentence “This is primarily due to the disordered component distribution leading to structural disorder, which affords infinite possibilities for tuning the polarization configuration” on page 3 has been updated to “This is primarily due to the disordered component distribution leading to unmatched atomic size, mass, valence state, and electronegativity, which induce random local strains and electric fields, providing infinite possibilities for tuning the local polarization configurations”.

2) “In high-entropy systems, the introduction of foreign ions with different properties enhances the local random field, which can disrupt the long-range order into small-sized PNRs, a phenomenon that provides for lowering the switching energy barrier as well as modulating the polarization configuration in polymorphic phase coexistence systems” has been added on page 7.

3) The sentence “This phenomenon reduces the polarization anisotropy^{42–44}, leading to the reduction of the domain switching energy barriers and thus ultimately achieving macroscopic SPE behavior” on page 8 has been updated to “The random distribution of local C-R-T-M-like phases indicates the decrease in polarization anisotropy and the existence of a strongly perturbed random field^{28,46–48}, which can effectively reduce the switching energy barriers and thereby lead to ideal macroscopic SPE behavior”.

4) The last paragraph of page 8 has been updated as follows: “The high-entropy effect results in significant differences in atom size, mass, charge state, and electronegativity, amplifying local structure disorder and causing random local fields. This phenomenon disrupts long-range ferroic order into locally diverse ferroic distortion with multiple BO_6 tilt types and rich heterogeneous configurations. The BO_6 tilt types can hinder the formation of electric field-induced long-range polarization. Locally interconnected C-R-T-M-like phases can drastically reduce the polarization anisotropy, thereby reducing the switching barrier, resulting in a flatter switching pathway and minimizing the hysteresis loss in the SPEs. Moreover, diverse polarization configurations can also increase the polarization direction and intensity, providing a strong polarization response”.

Comment 3: It is seen that the decrease in the polar cluster size in the RFE state is leading to the SPE state, any critical size (or its trigger, composition or temperature?) of this polar cluster that will lead to the SPE state? A discussion on this will be very helpful.

Response: Thank you for your suggestion. The critical size (or its trigger, composition, or temperature) that leads to the SPE state is discussed as follows: According to previous literature reports, the polar clusters of the SPE state are very small, usually a few nanometers. For size-driven SPEs, the SPE state occurs when the particle (grain or cluster) size (R) of the dielectric is in the range of $R_{\text{cr}} < R < R_{\text{c}}$ (where R_{cr} and R_{c} are the paraelectric limit and the correlation length of polarization fluctuations, respectively). Ideal size-driven SPEs consist of short-range-ordered nanometer polar clusters with a R of a few nanometers^{[Adv. Energy Mater. 10, 2001778 (2020)], [Appl. Phys. Lett. 124, 090501 (2024)]}. For temperature-driven SPEs, the SPE state is when the dielectric is in the region between T_{m} and T_{B} in the dielectric temperature spectrum (where T_{m} and T_{B} are the temperature corresponding to the maximum dielectric constant and Burns temperature, respectively)^{[Ferroelectrics 76, 241–267 (1987)], [Science 374, 100–104 (2021)], [Adv. Mater. 34, 2205787 (2022)]}. The main polar region in this state is only a few nanometers, which is comparable to the size of size-driven SPE particles. Therefore, the condition for obtaining the SPE state

at room temperature is to reduce T_m to room temperature or below and to induce small-sized polar clusters. Our specific ratio of BNBT to SLTT has a strong influence on the formation of polar clusters. The high entropy effect ($S_{\text{config}} = 1.61R$) due to a high SLTT content (SLTT-0.30 component) promotes the formation of smaller clusters and facilitates the desirable SPE state due to the presence of disorder and localized multiple symmetries. Current research on the energy storage performance of SPEs has focused on the temperature-driven type because it has a well-defined range of temperature boundaries (T_m to T_B). Our work is based on temperature-driven SPEs, so we have added a discussion of critical temperatures on page 3: Recently, superparaelectrics (SPEs) developed in RFEs have been considered as promising candidate materials for energy storage^{21–23}. The state of SPEs appears within the temperature range from T_m (the temperature corresponding to the maximum dielectric constant) to T_B (the Burns temperature) and is characterized by weakly coupled PNRs. Therefore, SPEs not only maintain a high maximum polarization (P_m) but also allow for flexible polarization redirection with small hysteresis, leading to a higher η compared to conventional RFEs. Also, on page 5 of the manuscript, there is a description of the T_m for SLTT-0.30 and SLTT-0.35: When S_{config} is increased to 1.61 (SLTT-0.30) and 1.66 R (SLTT-0.35), T_m drops below room temperature, indicating that both samples have reached room temperature SPE states.

Comment 4: The manuscript claims an extraordinary energy storage property of a bulk SPE composition, but the electrical performance characterizations have been carried out on samples of thickness 0.05 ± 0.01 mm, can it really be called a bulk sample? I am not sure.

Response: We appreciate the opportunity to clarify this aspect of our study. It can be determined that ceramics with a thickness of 0.05 ± 0.01 mm can be called bulk ceramics. We have taken care to align our terminology with that used in the broader literature; examples include Zhao et al., who described ceramics with a thickness of 0.035 mm as bulk ceramics, and similarly, Li et al., as well as Sun et al., whose reported bulk ceramics had thicknesses of 0.05–0.06 mm^[Nat. Commun. 14, 5725 (2023)], [Energy Environ. Sci.

16, 4511–4521 (2023)], [J. Am. Chem. Soc. 146, 13467–13476 (2024)]. These results are further supported by numerous reports that consistently refer to ceramics with thicknesses in the range of 0.03–0.06 mm as bulk ceramics [J. Am. Chem. Soc. 145, 19396–19404 (2023)], [J. Am. Chem. Soc. 146, 460–467 (2024)], [Small 19, 2206840 (2023)], [Mater. Horiz. 11, 1732–1740 (2024)]. With respect to the established conventions within our research community, we believe our use of the term “bulk” for ceramics measuring 0.04–0.06 mm in thickness is well justified.

Comment 5: Are the sample thicknesses similar in the performance comparison of energy storage ceramics shown in figure 3 (d) and (e), if not, can they really be compared?

Response: Thank you very much. In the performance comparison of the energy storage ceramics in Fig. 3 d and e, the sample thicknesses are not similar, ranging from 0.03–0.15 mm. However, this comparison is allowed. To the best of our knowledge, sample thickness is rarely emphasized in comparison plots of W_{rec} and η in articles on bulk ceramic energy storage [Science 384, 185–189 (2024)], [Nat. Commun. 13, 3089 (2022)], [Nat. Commun. 14, 5725 (2023)], [Energy Environ. Sci. 16, 4511–4521 (2023)], [Adv. Mater. 36, 2310559 (2024)], [Adv. Mater. 36, 2313285 (2024)], [J. Am. Chem. Soc. 146, 13467–13476 (2024)], [J. Am. Chem. Soc. 145, 19396–19404 (2023)], [J. Am. Chem. Soc. 146, 460–467 (2024)].

The comparison method in Fig. 3 d and e, that is, comparing the energy storage performance at their respective E_b , is adopted by most reports. This phenomenon is also present in recently published high-level journals. This is mainly due to the limitations of the preparation process. Most sintered ceramics usually need to be mechanically thinned and polished to meet the test requirements, which makes it difficult to accurately control the thickness changes during the thinning process. The thinning and polishing process can only be stopped by personal experience to determine whether the thickness meets the test requirements, and then the thickness can be measured using precise measuring instruments. As a result, this phenomenon leads to varying thicknesses of ceramics reported in the literature, making it difficult to compare them in a uniform manner. Perhaps in the future, a suitable standard will be proposed for energy storage ceramics, which can provide a more comprehensive method to compare the energy storage performance of different systems.

Comment 6: Is the electrical breakdown field, E_b , reported in the manuscript is the Weibull breakdown field? If not, better to report the Weibull E_b .

Response: Thank you for your suggestions. The E_b reported in the manuscript is directly measured using a ferroelectric test system rather than the E_b obtained by Weibull distribution. In order to verify the high reliability of E_b , we have carried out a Weibull distribution validation of the SLTT-0.30 high-entropy SPE according to your suggestion, as shown in Supplementary Fig. 11. Our total number of samples is 10, and the obtained Weibull modulus β is 19.8, indicating high reliability. The theoretical E_b is 725.6 kV cm^{-1} , which is very close to the experimental E_b value (710 kV cm^{-1}). We have added a description of this on page 10 of the manuscript: The reliability of E_b is verified using Weibull distribution analysis, as shown in Supplementary Fig. 11. The Weibull modulus (β) of SLTT-0.3 sample is 19.8, indicating that the sample possesses high reliability and homogeneity⁵². The calculated statistical E_b is 725.6 kV cm^{-1} , which is very close to the E_b obtained in the P - E loop.

Supplementary Fig. 11 Weibull distribution of theoretical E_b for SLTT-0.30 ceramic.

Reviewer #3

Remarks to the Author: The capacitive energy storage is in an area of current research and interest. Recently, more and more attention is paid to both ultrahigh W_{rec} and η , especially for the lead-free bulk ceramics. This work demonstrates that the $(1-x)\text{BNBT}-x\text{SLTT}$ presents a superior comprehensive energy storage performance due to the entropy engineering strategy, which disrupt long-range ferroic orders into local polymorphic distortion disorder and rich heterogeneous polarization configurations. The high-entropy superparaelectrics (SLTT-0.30) exhibits a large W_{rec} of 15.48 J cm^{-3} and an ultrahigh η of 90.02% at 710 kV cm^{-1} . The manuscript is well written, and the findings are important to the broader ferroelectrics community. However, there are some issues to be addressed. My comments and concerns are provided as follows:

Response: We are very grateful for your careful review and positive comments. In our work, we demonstrate that entropy engineering is an advanced approach to designing ideal bulk SPEs through advanced characterization. This design induces local ferroic distortion, which reduces the switching barrier and polarization anisotropy, and leads to excellent energy storage performance of $W_{\text{rec}} \sim 15.48 \text{ J cm}^{-3}$ and $\eta \sim 90.02\%$ in high-entropy bulk SPEs. Through our in-depth analysis, we believe that this can provide a valuable basis for improving energy storage performance and is worth promoting. We have responded to each of your comments and have added relevant content to the manuscript, which has significantly improved the quality of our manuscript.

Comment 1: In terms of “High-entropy”, it is a hot topic in recent. Does more element doping or solid solution mean high entropy? Can the superparaelectrics only be attributed to the high-entropy systems?

Response: Thank you very much. With more doping elements or solid solutions, the entropy gets higher. There are currently two main definitions of high-entropy: one based on components and one based on configuration entropy (S_{config}). In the former case, a high-entropy material contains more than five major metallic elements, and the molar concentration of each element is between 5% and 35%^[Acta Mater. 122, 448–511 (2017)]. The

latter depends on the value of S_{config} . S_{config} can be calculated using the following equation^[Nat. Energy 8, 956–964 (2023)]:

$$S_{\text{config}} = -R \left(\sum_{i=1}^N x_i \ln x_i + \sum_{j=1}^M x_j \ln x_j \right) \quad (6)$$

where R , N (M) and x_i (x_j) are the ideal gas constant, atomic species and contents at the equivalent cation (anion) sites, respectively. The judgment of entropy in this work comes from the value of S_{config} . Therefore, according to the equation, the entropy increases as the number of elements increases. However, there is a special case where the entropy for a specific number of elements reaches a maximum when the number of elements is high and the atomic fractions are equal^[Mater Sci Eng R Rep. 146, 100644 (2021)].

Superparaelectric (SPEs) are not only attributed to high-entropy systems. For example, the NaNbO_3 -based ceramics prepared by Liu et al., the AgNbO_3 -based ceramics prepared by Liao et al., and the BaTiO_3 -based ceramics prepared by Sun et al. are not high-entropy ceramics, although they are SPEs^[Microstructures 3, 2023009 (2023)],^[Chem. Eng. J. 448, 150901 (2024)],^[J. Am. Chem. Soc. 145, 6194–6202 (2023)]. In addition, in some SPE films, such as $(\text{Ba}_{0.95}, \text{Sr}_{0.05})(\text{Zr}_{0.2}, \text{Ti}_{0.8})\text{O}_3$ films prepared by Wang et al.^[Adv. Energy Mater. 10, 2001778 (2020)] and (Sm)-doped BiFeO_3 - BaTiO_3 films prepared by Pan et al. are not high-entropy systems^[Science 374, 100–104, (2021)]. Our work is based on the correlation between the two and proves that the high entropy strategy is a feasible strategy for inducing ideal SPEs.

Comment 2: The authors claim that the inverse relationship between W_{rec} and η in Introduction part. It is unclear where that comes from. Many factors are not certain to be related to the W_{rec} and η . The authors should give more comments. It would help reader understanding the relationship between W_{rec} and η .

Response: We greatly appreciate your careful review. In many reports, such as the NaNbO_3 (NN)-based and AgNbO_3 (AN)-based systems, the existence of an AFE-FE phase transition usually leads to lower efficiency. For example, the W_{rec} of NN-based ceramics prepared by Jiang et al. is 18.5 J cm^{-3} and η is 78.7%^[Energy Storage Mater. 43, 383–390 (2021)]. The W_{rec} of AN-based ceramics prepared by Chen et al. is 11.4 J cm^{-3} and η is 80%^[Small 20, 2306486 (2023)]. In linear dielectrics, although high η can be obtained, their

relatively low P_m usually leads to a low W_{rec} . This situation often occurs in ceramics such as SrTiO₃-based and CaTiO₃-based. In addition, in many systems, as the electric field increases, P_r and hysteresis losses usually increase, which will cause η to decrease^{[Nano Energy 109, 108275 (2023)], [J. Mater. Chem. C 7, 14384–14393 (2019)], [Small 19, 2206662 (2023)]}. Although W_{rec} increases with the electric field. Therefore, the inverse relationship between W_{rec} and η comes from the above statement, which is also a relationship mentioned in many articles^{[J. Am. Chem. Soc. 145, 11764–11772 (2023)], [J. Am. Chem. Soc. 145, 6194–6202 (2023)]}. To help readers understand the relationship between W_{rec} and η , we have added the following discussion in the second paragraph of the **Introduction** part (page 2):

1) “their η are capped at or below 80%, which is mainly caused by the AFE-ferroelectric (FE) phase transition”.

2) “such as CaTiO₃ (CT)-based and SrTiO₃ (ST)-based ceramics^{15–18}, the relatively low intrinsic polarization leads to their W_{rec} usually being lower than 7 J cm⁻³. In addition, the maximum polarization (P_m) rises with the increase in electric field, thus contributing to the enhancement of W_{rec} . However, this process is also accompanied by an increase in remnant polarization (P_r), hysteresis loss, and leakage current, all of which have negative impacts on η ^{19,20}. Therefore, the trade-off between W_{rec} and η has become a primary challenge in designing high-performance dielectric ceramics”.

Comment 3: It is not clear that high-entropy systems correspond to low free energy, potentially resulting in reduced barriers. Clarify in the text.

Response: Thank you for your suggestion. According to your reminder, after an in-depth literature survey, we believe that the statement mentioned in the manuscript that high-entropy systems correspond to low free energy is not very rigorous. Therefore, after comprehensive consideration, we decided to delete this sentence, which has no impact on the overall logic and research focus of the manuscript. The thermodynamic relationship between free energy (ΔG) and entropy (ΔS) in high-entropy systems is:

$$\Delta G = \Delta H - T\Delta S$$

where ΔH is the enthalpy and T is the temperature. Based on this equation, Yang et al. mentioned that entropy increases with the introduction of multiple elements, which

causes ΔG to decrease^[Nat. Mater. 21, 1074–1080 (2022)]. Liu et al. calculated that the free energy profile of high-entropy ferroelectrics is smoother than that of normal ferroelectrics, based on the Landau-Devonshire theory^{[Ceram. Int. 47, 33039–33046 (2021)], [InfoMat 5, 12488 (2023)]}. However, the effect of ΔH should not be neglected according to the thermodynamic relation equation. In some cases, if $\Delta S \geq 1.5 R$, this can lead to $T\Delta S$ being large enough to dominate the free energy landscape and overcome ΔH ^{[Mater Sci Eng R Rep. 146, 100644 (2021)] [Prog. Mater. Sci. 145, 101300 (2024)]}. This is more pronounced at high temperatures. Nevertheless, there are also cases, such as at low temperatures, where the entropy gain may not be sufficient to overcome ΔH ^{[Acta Mater. 122, 448–511 (2017)], [Adv. Mater. 31, 1806236 (2019)]}. Therefore, we believe that it is not very rigorous to say that high-entropy systems correspond to low free energy.

Comment 4: A key challenge of this work is that SLTT-0.30 is added into BNBT to regulate configuration entropy. This solid solution content is relatively large to the ceramic. How to ensure the absence of impurities and uniform distribution of multiple elements.

Response: Thank you so much. In high-entropy systems, the sluggish diffusion effect can improve phase stability in addition to inhibiting grain growth. Specifically, the strong lattice distortion caused by high entropy can increase the energy barrier for atomic diffusion (sluggish diffusion effect), which can effectively delay phase separation and elemental segregation and thus improve phase stability and uniformity of elemental distribution^{[Mater Sci Eng R Rep. 146, 100644 (2021)], [Nat. Commun 12, 5747 (2021)]}. In addition, before selecting the dopant and doping amount, we investigated the literature related to BNT-based ceramics, such as in $(1 - x)(0.94\text{Na}_{0.5}\text{Bi}_{0.5}\text{TiO}_3 - 0.06\text{BaTiO}_3) - x\text{Ca}_{0.7}\text{La}_{0.2}\text{TiO}_3$ (x taking values up to 0.43), $(1 - x)\text{Bi}_{0.5}\text{Na}_{0.5}\text{TiO}_3 - x\text{Sr}_{0.7}\text{La}_{0.2}\text{Zr}_{0.15}\text{Ti}_{0.85}\text{O}_3$ (x takes values up to 0.42), $(1 - x)[0.955(\text{Bi}_{0.5}\text{Na}_{0.5})\text{TiO}_3 - 0.045\text{Ba}(\text{Al}_{0.5}\text{Ta}_{0.5})\text{O}_3] - x\text{CaTiO}_3$ (x takes values up to 0.4), $(1 - x)(\text{Bi}_{0.47}\text{La}_{0.03}\text{Na}_{0.5})_{0.94}\text{Ba}_{0.06}\text{TiO}_3 - x\text{SrTi}_{0.875}\text{Nb}_{0.1}\text{O}_3$ (x taking values up to 0.40), $(1 - x)(0.75\text{Na}_{0.5}\text{Bi}_{0.5}\text{TiO}_3 - 0.25\text{SrTiO}_3) - x\text{CaTi}_{0.875}\text{Nb}_{0.1}\text{O}_3$ (x taking values up to 0.5), $(1 - x)(0.94\text{Na}_{0.5}\text{Bi}_{0.5}\text{TiO}_3 - 0.06\text{BaTiO}_3) - x\text{CaTi}_{0.8}\text{Hf}_{0.2}\text{O}_3$ (x taking values up to 0.3), $(1 -$

x) $\text{Bi}_{0.5}\text{Na}_{0.5}\text{TiO}_3$ - $x\text{Sr}_{0.7}\text{Bi}_{0.2}\text{Ti}_{0.8}\text{Hf}_{0.2}\text{O}_3$ (x taking values up to 0.4) and other systems, there are no second phases, although all of them are doped with larger dopant amounts [Adv. Funct. Mater. 33, 2301027 (2023)], [Chem. Eng. J. 446, 137105 (2022)], [J. Mater. Chem. A 10, 9535–9546 (2022)], [J. Materiomics 8, 537–544 (2022)], [Appl. Mater. Interfaces 14, 54051–54062 (2022)], [Compos. B. Eng. 255 (2023) 110630], [Chem. Eng. J. 455, 140924 (2023)]. This phenomenon indicates that the BNT-based ceramics have a large solid solubility. The following aspects should also be taken into account to ensure that the elements are uniformly distributed and impurity-free during the experimental process:

1) The use of high-purity raw materials can minimize the introduction of impurities.

2) The use of high-precision weighing equipment can improve the accuracy of chemometrics. It is also necessary to prevent the loss of raw materials during the preparation process.

3) During ball milling, high-energy ball milling and increased ball milling duration can be utilized to apply sufficient energy to improve the uniformity of elemental distribution and facilitate the reaction. Moreover, the ratio of raw materials, zirconia balls and media should be set reasonably to improve ball milling efficiency.

4) A reasonable sintering temperature gradient and holding time gradient should be set. On the one hand, it can ensure the full reaction, and on the other hand, it can prevent elemental volatilization or secondary grain growth.

Finally, according to your question, we think that XRD tests should be added to describe the phase structure of SLTT- x ceramics. As shown in Supplementary Fig. 3, all components exhibit a typical perovskite structure without a secondary phase.

Supplementary Fig. 3 XRD patterns of the SLTT- x ceramics.

Comment 5: The authors use the HR-TEM results to demonstrate the absence of the large-scale ferroelectric domain. Did the authors check the lower-magnification TEM results? More TEM images should be provided in SI.

Response: Thank you for your suggestion. In fact, we checked the low-magnification TEM images shown in Supplementary Fig. 6. However, we neglected the importance of low-magnification TEM for detecting large-scale domains and therefore did not present low-magnification TEM in this work. To compensate for this omission, we have added the low-magnification TEM in the Supplementary Information. As shown in Supplementary Fig. 6, the TEM images again show no large-scale domain morphology, which is consistent with the results observed by HR-TEM. In addition, we have also revised the relevant content on page 6 of the manuscript.

Supplementary Fig. 6 Low-magnification TEM images along the **a** $[100]_c$ and **b** $[110]_c$ directions.

Comment 6: Fig. 2f shows the PFM image after poling treatment with ± 30 V of SLTT-0.30 ceramic and no domain switching was found. But why? Authors state that high-entropy may give rise to the low free energy, potentially resulting in reduced domain switching barriers. The pinning effect of defects maybe considered. Please elaborate this observation/result further.

Response: Thank you for your question. The reason that no domain switching is found in SLTT-0.30 may be related to the reduction of domain size. In SPE high-entropy systems, the introduction of foreign ions with different properties enhances the local random field, thus disrupting the large-size domains into small-size PNRs, which has been confirmed in TEM. This phenomenon helps to enhance the weak coupling and dynamic properties of PNRs. These results can explain the absence of domain switching

in the PFM of SLTT-0.30. On the one hand, the weak coupling property of small-sized PNRs in SPE can delay the formation of large-sized field-induced domains. Therefore, only small-size domains are induced at a voltage of ± 30 V, and the limited resolution of the PFM makes it difficult to detect the small-size domains induced at this voltage. On the other hand, the highly dynamic nature of PNRs causes the field-induced domains to quickly return from the induced state to the initial state after unloading the voltage. The PFM images were recorded after the voltage was unloaded, at which time the field-induced domains may have recovered to their initial state, resulting in the failure to detect domain switching in time. To explain this phenomenon more clearly, we have added an explanation of Supplementary Fig. 7 in the Supplementary Information: “In contrast, no domain switching is detected in the SLTT-0.30 high-entropy SPE. This phenomenon may be related to the formation of PNRs. On the one hand, the weak coupling characteristics of PNRs in SPEs delay the formation of large-size field-induced domains. Therefore, only small-size domains are induced at a voltage the ± 30 V, and the limited resolution of PFM makes it difficult to detect the small-size domains induced under the condition. On the other hand, the high dynamic characteristics of PNRs cause the field-induced domains to recover rapidly from the induced state to the initial state after unloading the voltage. The PFM images are recorded after voltage unloading, so the domain switching cannot be detected in time”.

As for the statement that high entropy corresponds to low free energy, we think it needs to be further proven. This is because, according to the thermodynamic equations, the free energy is affected by enthalpy in addition to entropy, which we have explained in detail in **Comment 3**. However, the claim that high entropy leads to lower switching energy barriers in high-entropy systems should be reasonable and can be proved in a large number of studies. From the thermodynamic point of view, the decrease in domain size leads to a decrease in the domain switching energy barrier^[Science 374, 100–104 (2021)], [Appl. Phys. Lett. 124, 090501 (2024)]. The high-entropy strategy is an effective method that can break the long-range ordering and reduce the domain size^[Nat. Energy 2023, 8, 956], [Nano-Micro Lett. 15, 65 (2023)]. In addition, high entropy can induce the coexistence of multiple local symmetries, which can reduce the polarization anisotropy as well as further reduce the domain

switching barrier, resulting in flatter switching paths and minimizing hysteresis losses^{[Science 384, 185–189 (2024)], [Acta Mater. 236, 118115 (2022)], [Nano Energy 104, 107910 (2022)], [Nat. Commun. 13, 3089 (2022)]}. As a result, we have achieved SPE behaviors with ideal polarization forms at the macroscopic level.

Undoubtedly, the formation of defects such as oxygen vacancies usually causes a pinning effect on the domain walls, hindering the movement of the domain walls or the reorientation of the PNRs^{[J. Am. Ceram. Soc. 105, 6479–6507 (2022)] [Research, 2022, 9764976 (2022)]}. However, the P - E loop of SLTT-0.30 at a low electric field shows that it has a linear response with low P_r (see Supplementary Fig. 2b). It suggests that the domains in SLTT-0.30 possess high dynamics and are able to switch quickly under the electric field and recover quickly after unloading the electric field. Therefore, we believe that the pinning effect produced by the defects is weak.

Reviewer #4

Remarks to the Author: The paper reports on the fabrication of dielectric capacitors capable of simultaneously achieving high energy storage density and high charge-discharge efficiency by inducing an increase in configurational entropy in SPE states through the design of (1-x)BNBT-xSLTT composition. The authors aptly demonstrate the existence of highly dynamic PNRs in T, R, M-like phases through TEM, PFM, and Raman analysis of sintered ceramics. Furthermore, they logically correlate the superior properties of high W_{rec} and efficiency in the SLTT-0.30 specimen with the presence of highly dynamic PNRs in various phases. Based on the excellent energy storage performance data, the paper appears to be quite well-written.

However, from the perspective of this journal's requirements, the strategy of energy storage performance enhancement through entropy engineering is not entirely novel. While it's true that energy storage density and efficiency are important performance metrics for dielectric capacitors, it is now crucial to demonstrate the practical implementation of energy storage and utilization of dielectric capacitors. Therefore, I urge the authors to address and incorporate the following comments into the manuscript.

Response: We sincerely appreciate your careful review and point out that we have logically linked the polymorphic PNRs and the excellent energy storage performance. We are encouraged by this. Regarding your concern about innovation, we explain as follows:

As you mentioned, entropy engineering has been widely proven to enhance dielectric energy storage performance, for instance, in bulk ceramics, multilayer ceramic capacitors, and thin films. However, there is no report on the simultaneous realization of the combined performance of $W_{\text{rec}} > 15.48 \text{ J cm}^{-3}$ and $\eta > 90\%$ in a single high-entropy bulk ceramic or a single SPE bulk ceramic. In other reports, the combination of high W_{rec} and high η is also rarely achieved. Our research is based on the thermodynamic relationship between high entropy and SPE to improve energy storage performance and deeply explore the internal mechanism.

The structural disorder induced by high entropy can lead to diverse small-sized local configurations rather than a single or simple two-phase coexistence, which minimizes the energy barrier required for domain switching^{[Science 100–104 (2021)], [Nanoscale 12, 19582–19591 (2020)]}. Coincidentally, this is exactly what is needed for an ideal SPE state, achieving a low-hysteresis polarization reorientation process by reducing the switching energy barrier^[Appl. Phys. Lett. 124, 090501 (2024)]. Therefore, we combined high entropy and SPE to promote energy storage performance through entropy-tailored, local ferroic distortion. First, the induction of local heterogeneous polarization configurations, including R, T, and M-like configurations embedded in the C matrix, reduces the domain size and polarization anisotropy. This leads to a reduction in the switching energy barrier, resulting in SPE behavior with ideal polarization forms. Second, the multiple BO₆ tilt types caused by high entropy can significantly delay the polarization saturation of SPE. Finally, based on these synergistic effects, we achieve an outstanding W_{rec} of 15.48 J cm⁻³ and an ultrahigh η of 90.02% simultaneously in high-entropy SPEs under a large E_b of 710 kV cm⁻¹. This represents the best comprehensive energy storage performance reported so far among bulk SPEs.

We have explored the connection between design strategy, local structure and energy storage performance, and proved that entropy engineering is an effective method for designing high-performance SPE. Therefore, we believe that our work is worth promoting. It is important that we have responded to your questions and suggestions and revised the manuscript accordingly, which is very helpful to improve the quality of our manuscript. We hope that these responses and revisions can address your concerns.

Comment 1: Practicality of Energy Density Values

In evaluating the energy density (W_{rec}) of the SLTT-0.30 specimen, the authors utilized samples with very thin thickness and small electrode areas. It cannot be guaranteed that high W_{rec} will be maintained as the electrode area and sample thickness increase. For instance, if the sample thickness is at the level of 0.3 to 0.5 mm and the electrode diameter is at the level of 3 to 5 mm, will high E_b and W_{rec} be sustained? (Indeed, many papers use thicknesses and electrode areas even larger than these.) It would be

beneficial to demonstrate the behavior of E_b and W_{rec} concerning sample thickness.

Response: Thank you very much. As you mentioned, sample thickness and electrode size have an effect on E_b and W_{rec} . In particular, smaller thickness does favorably improve E_b , which is associated with a reduced probability for the occurrence of defects such as pores, voids, or microcracks^[Chem. Rev. 121, 6124–6172 (2021)]. However, among the factors affecting E_b , in addition to thickness, lattice distortion, grain size, dielectric loss, band gap, leakage current and other factors also play an important role in E_b , which we have analyzed in detail in the last paragraph of page 9 of the manuscript and in Supplementary Fig. 9 and 10. In addition, from the perspective of practical applications such as device miniaturization and integration, the size of ceramic capacitors is usually required to be made smaller. For example, in the reported multilayer ceramic capacitors, the thickness of the single-layer ceramic is in the range of 1 to 10 μm , which is much lower than the thickness reported in our work^[Science 384, 185–189 (2024)],^[Nat. Commun. 14, 1166 (2023)],^[Energy Environ. Sci. 13, 4882–4890 (2020)],^[Adv. Energy Mater. 14, 2304291 (2024)],^[Adv. Energy Mater. 2400821 (2024)]. Our sample thicknesses and electrode sizes were set based on recently published high-level articles, primarily to enable equal comparisons with the energy storage performance in these journals. In fact, ceramics characterized by high density and few defects can be thinned without breaking, while low-quality ceramics are difficult to thin. In articles on dielectric energy storage, when presenting and comparing W_{rec} , E_b , and η , the sample thickness is usually not emphasized^[Science 384, 2024, 185],^[Nat. Commun. 2022, 13, 3089],^[Nat. Commun. 2023, 14, 5725],^[Energy Environ. Sci. 2023, 16, 4511],^[Adv. Mater. 36, 2024, 2310559],^[Adv. Mater. 2024, 36, 2313285],^[J. Am. Chem. Soc. 2024, 146, 13467],^[J. Am. Chem. Soc. 2023, 145, 19396],^[J. Am. Chem. Soc. 2024, 146, 460]. Instead, the energy storage performance at each E_b is presented and compared, as shown in Fig. 3 in the manuscript.

Table R3 lists the thickness (t) and electrode diameter (d) of energy storage ceramics reported in recent high-level articles. It can be seen that our ceramic thickness and electrode diameter are almost exactly the same as or even larger than those reported in these journals.

Table R3 Parameters of ceramics recently reported in high-level articles. (BT: BaTiO₃; BNT: Bi_{0.5}Na_{0.5}TiO₃; BF: BiFeO₃; KNN; K_{0.5}Na_{0.5}NbO₃; ST: SrTiO₃)

System	Reference	t (mm)	d (mm)
BT	[Science 384, 185–189 (2024)]	0.05–0.1	1
BNT	[J. Am. Chem. Soc. 146, 13467–13476 (2024)]	0.05–0.06	0.5

BNT	[Nat. Commun. 14, 5725 (2023)]	0.035	/
BNT	[J. Am. Chem. Soc. 146, 460–467 (2024)]	0.05	1
BNT	[Energy Environ. Sci. 16, 4511–4521 (2023)]	0.05	2
BT	[Adv. Mater. 36, 2313285 (2024)]	0.06–0.08	1
BNT	[J. Am. Chem. Soc. 145, 19396–19404 (2023)]	0.05–0.07	1
BNT-BT-NN	[Adv. Mater. 34, 2205787 (2022)]	0.05–0.07	1.38
NN	[Small, 19, 2303915 (2023)]	0.05–0.07	1
NN	[Mater. Today Phys. 38, 101208 (2023)]	0.035	2
KNN	[Nat. Commun. 13, 3089 (2022)]	0.06–0.1	1
ST	[Mater. Horiz. 11, 1732–1740 (2024)]	0.03	/
This work	This work	0.04–0.06	1.5

Comment 2: Implementation of Energy Charge-Discharge Performance in Practical Devices

Building upon the preceding comment, it would be beneficial if the paper could introduce, even in a simple form, the implementation of a device utilizing the capacitor of SLTT-0.30 composition for energy charge-discharge applications. For instance, demonstrating the operation of a simple device driven by stored energy would be advantageous.

Response: Thank you very much. We agree that demonstrating the practical application of energy storage ceramics is of great interest and a goal that should be pursued by all researchers in the field. However, achieving this is a major challenge due to the requirement for highly professional equipment and techniques. To realize the charge-discharge applications in real devices, it is typically necessary to prepare multilayer ceramic capacitors (MLCCs), which have wide applications in pulsed power devices and systems. The MLCCs are fabricated by a series of processing steps, which include slurry preparation, tape-casting, screen printing, lamination, co-sintering, and termination, as shown in Figure R1. This indicates that the preparation process is complicated and requires very professional equipment and technology, as well as rich experience.

The single-layer ceramics prepared by Li et al. exhibited a W_{rec} of 2.5 J cm^{-3} and

η of 95%, with the MLCCs made from this composition demonstrating a W_{rec} of 9.5 J cm^{-3} and η of 92%^[Adv. Mater. 30, 1802155 (2018)]. Ji et al. obtained a W_{rec} of 7.5 J cm^{-3} and η of 92% for single-layer ceramics, while the MLCCs reached a W_{rec} of 18 J cm^{-3} and η of 93%^[Energy Stor. Mater. 38, 113–120 (2021)]. Zhang et al. achieved W_{rec} of 10 J cm^{-3} and η of 93.5% for single-layer ceramics, with MLCCs showing W_{rec} of 20.8 J cm^{-3} and η of 97.5%^[Science 384, 185–189 (2024)]. In our work, we reported that SLTT-0.30 ceramic achieved a W_{rec} of 15.48 J cm^{-3} and η of 90.02%. Obviously, if the SLTT-0.30 component is developed into MLCCs, the energy storage performance can be further improved.

We all agree that demonstrating the drive of energy storage has great benefits, but we are currently facing great difficulties in this regard due to laboratory conditions, research directions and time constraints. In addition, to demonstrate energy drive performance on devices, more knowledge may be required, such as circuit design, simulation and manufacturing. In the current reports on energy storage ceramics, there is no relevant research that can be used as a reference. Our main goal is to find a strategy that can significantly improve the energy storage performance of ceramics, and to fundamentally clarify the relationship between this strategy and energy storage performance, so as to provide a basis for the subsequent development of high-performance ceramics. This is also the research idea behind most of the reports related to energy storage ceramics. To achieve energy drive at the device level, we still need to keep trying, and this is also the direction we must work towards in the future.

[Redacted]

Figure R1 The schematic diagram of the MLCCs fabrication process^[Chem Rev,121, 6124 (2021)].

Comment 3: How was the E_b data in Fig. 3c determined? How many specimens were measured to determine E_b ? How is the reproducibility of the energy storage

performance?

Response: Thank you for your question. The E_b in Fig. 3c was obtained from the P - E loop tested by the ferroelectric test system (see Fig. 3a). E_b can be easily determined using very few samples. This is mainly because the SLTT-0.30 sample, in addition to the lattice distortion and high resistivity caused by high entropy, also has the advantages of high quality, small grain size, dense microstructure, high bandgap width, low dielectric loss, etc., which are beneficial to improving E_b . In order to verify the reproducibility of energy storage performance, we retested the P - E loops of five SLTT-0.30 samples and calculated the corresponding W_{rec} and η . As shown in Figure R2, the W_{rec} and η of each sample show acceptably small fluctuations. These results show the high reproducibility and reliability of energy storage performance.

To further verify the reliability of E_b , we added Weibull distribution analysis, as shown in Supplementary Fig. 11. The Weibull modulus (β) obtained by fitting is 19.8, which indicates a high reliability and homogeneity of the ceramics^[Science 384, 185–189 (2024)][Nat Commun 14, 5725 (2023)]. The calculated statistical E_b is 725.6 kV cm^{-1} , which is very close to the E_b of 710 kV cm^{-1} in Fig. 3a and c. These results further demonstrate that E_b has high reproducibility. To improve the reliability of E_b in the manuscript, we added the Weibull analysis plot in the Supplementary Information and added the following explanation on page 10 of the manuscript: The reliability of E_b is verified using Weibull distribution analysis, as shown in Supplementary Fig. 11. The Weibull modulus (β) of SLTT-0.3 sample is 19.8, indicating that the sample possesses high reliability and homogeneity⁵². The calculated statistical E_b is 725.6 kV cm^{-1} , which is very close to the E_b obtained in the P - E loop.

Figure R2. P - E loops of different samples at their respective E_b .

Supplementary Fig. 11 Weibull distribution of theoretical E_b for SLTT-0.30 ceramic.

Comment 4: It would be beneficial to add the sources of the comparison data in Fig. 3d and e to the supplementary information.

Response: Thank you for your suggestion. Indeed, adding the source of the comparison data in Fig. 3d, e will help readers to improve the credibility of the performance comparison. This will also make it easier for readers to track the literature. Therefore, we followed your suggestion and added the corresponding data sources in the Supplementary Information., as shown in Supplementary Table 2. The corresponding references have been added to the “References” section in the Supplementary

Information.

Supplementary Table 2 Relevant references for the Fig. 3d, e.

Figure	Reference
Fig. 3d, 3e	4, 10, 9, 11, 22–96

Comment 5: For the fabrication of (1- x)BNBT- x SLTT ceramics, the authors separately calcined BNBT and SLTT before mixing them again for sintering. Is there any specific reason for not simultaneously mixing all raw materials and calcining them? Are there no secondary phases present in the sintered (1- x)BNBT- x SLTT ceramics? It would be beneficial to add XRD data and lattice information obtained through refinement of the sintered (1- x)BNBT- x SLTT ceramics to the supplementary information.

Response: Thank you very much. Separate calcination of the dopant and base material is a common calcination method in the solid-phase reaction^{[Nano Energy 101, 107577(2022)],[J. Materiomics 8, 489–497 (2022)][ACS Appl. Mater. Interfaces 14, 17652–17661 (2022)],[J. Mater. Chem. A, 10, 9535–9546 (2022)]}. Previous studies have shown that ceramics prepared using this method usually exhibit a large breakdown electric field (E_b). For instance, ceramics prepared by Chen et al. achieved an E_b of 800 kV cm⁻¹, and ceramics prepared by Wu et al. achieved an E_b of 646 kV cm⁻¹[J. Mater. Chem. A 9, 4789–4799 (2021)],[Small 19, 2303915 (2023)]. Therefore, we separately calcined BNBT and SLTT before mixing them again for sintering.

We have neglected the importance of XRD for the detection of phase structure. Thanks to your reminder, we have remeasured the XRD data and carried out structural refinement on (1- x)BNBT- x SLTT. The corresponding results are shown in Supplementary Fig. 3 and 4. The results indicate that the samples demonstrate a typical perovskite structure without a secondary phase, characterized by the coexistence of R and T phases. We have added a corresponding explanation on page 6 in the manuscript: “X-ray diffraction (XRD) reveals that both SLTT- x ceramics possess a typical perovskite structure without impurities (Supplementary Fig. 3). The refinement results and phase compositions obtained by Rietveld refinement via GSAS-II software are shown in Supplementary Fig. 4³³. SLTT- x ceramics exhibit a phase structure in which the R and T phases coexist. As x increases, the proportion of R phase decreases and the

proportion of T phase increases”.

Supplementary Fig. 3 XRD patterns of the SLTT-*x* ceramics.

Supplementary Fig. 4 a–e Rietveld refinement results of XRD patterns obtained by GASA-II software for SLTT-*x* ceramics. f Phase proportions of SLTT-*x* ceramics obtained by refinement.

Comment 6: Page 11, “Structural stability is also associated with reduced crystal structure symmetry caused by polymorphic PNRs.”: If there could be further elaboration on this sentence, it would be beneficial.

Response: We sincerely appreciate your suggestion. Studies have shown that polymorphic PNRs in high-entropy systems, especially continuous intermediate polarization states, can effectively break the constraints of crystallographic symmetry.

This can significantly reduce the polarization anisotropy, thereby effectively facilitating polarization rotation between multiphases and lowering the switching barriers, leading to improved stability^{[Nano Energy 112, 108458 (2023)], [Acta Mater. 236, 118115 (2022)]}. However, we admit that the statement “Structural stability is also associated with reduced crystal structure symmetry caused by polymorphic PNRs” is not direct enough and may be misleading. What we want to make clear is that polymorphic PNRs can improve temperature and frequency stability. Because in a system with polymorphic PNRs, the reduction in polarization anisotropy can reduce the switching barrier, which is conducive to a flatter switching path. Moreover, the decrease in domain size can enhance the weak coupling effect. Therefore, these small-sized PNRs can respond quickly to external electric fields, thereby reducing hysteresis losses and improving temperature and frequency stability under applied electric fields. To this end, we made the following revision on page 12 of the manuscript: “the entropy-induced polymorphic PNRs weaken the inter-coupling effect, lower the switching barrier, and promote the polarization rotation. Thus, these PNRs can respond quickly to the applied electric field, leading to enhanced stability^{44,52}. Benefiting from these synergistic effects, the energy storage performance of SLTT-0.30 ceramic demonstrates excellent stability at different temperatures and frequencies”.

Comment 7: In Fig. S8, the T_m of the SLTT-0.30 specimen appears to be higher than 50 °C. This contradicts the results shown in Fig. S1d. Additionally, there is a typo “date” in Fig. S8 that needs to be corrected.

Response: Thank you for your careful review. The curve in the figure was obtained at 1000 kHz rather than 1 kHz, thus resulting in a T_m higher than room temperature, which was an oversight on our part. We have corrected the curve to that at 1 kHz and then re-fitted the T_B and pointed out T_m and T_B in the figure as shown in Supplementary Fig. 12 (Fig. S8 mentioned in the comment 7). In addition, we have corrected the incorrect word “date” to “data” in this figure. Finally, we have checked all images as well as the manuscript content to avoid errors.

Supplementary Fig. 12 Temperature dependence of the reciprocal of ϵ_r at 1 kHz to determine T_B for SLTT-0.30 ceramic.

REVIEWERS' COMMENTS

Reviewer #1 (Remarks to the Author):

I agree with the corrections the authors have made to the issues that were raised in the previous review. I recommend acceptance.

Reviewer #2 (Remarks to the Author):

The comments raised by me are satisfactorily addressed and included in the revised manuscript. I am ok with the revision. The modified manuscript may be accepted for publication.

Reviewer #4 (Remarks to the Author):

The authors appear to have adequately addressed the issues I raised. I recommend the publication of this work.

Response Letter to Manuscript “High-entropy superparaelectrics with locally diverse ferroic distortion for high-capacitive energy storage” (NCOMMS-24-22221B)

Response to Reviewer’s Comments

We would like to thank the reviewers again for their valuable time and effort in reviewing our manuscript. We also sincerely thank the reviewers for their positive comments on the manuscript and for recommending it for publication.

Reviewer #1

Remarks to the Author: I agree with the corrections the authors have made to the issues that were raised in the previous review. I recommend acceptance.

Response: Thank you for confirming that our corrections have addressed your issues and recommending the manuscript for acceptance.

Reviewer #2

Remarks to the Author: The comments raised by me are satisfactorily addressed and included in the revised manuscript. I am ok with the revision. The modified manuscript may be accepted for publication.

Response: We are very grateful to you for being satisfied with the revised manuscript and recommending it for publication.

Reviewer #4

Remarks to the Author: The authors appear to have adequately addressed the issues I raised. I recommend the publication of this work.

Response: We appreciate you pointing out that we have adequately addressed your issues and recommending our work for publication.